# The Rheb GTPase promotes pheromone blindness via a TORC1-independent pathway in the phytopathogenic fungus *Ustilago maydis*

Antonio de la Torre[1¤], José Pérez-Martín[1,2,3]*

**1** Instituto de Biología Funcional y Genómica (CSIC), Salamanca, Spain, **2** Instituto de Biomedicina de Valencia (CSIC), Valencia, Spain, **3** Instituto de Biología Molecular y Celular de Plantas (CSIC-UPV), Valencia, Spain

¤ Current address: Centro Nacional de Biotecnología (CSIC), Madrid, Spain
* jose.perez@csic.es

**Data Availability Statement:** All relevant data are within the manuscript and its Supporting Information files.

## Abstract

The target of the rapamycin (TOR) signaling pathway plays a negative role in controlling virulence in phytopathogenic fungi. However, the actual targets involved in virulence are currently unknown. Using the corn smut fungus *Ustilago maydis*, we tried to address the effects of the ectopic activation of TOR on virulence. We obtained gain-of-function mutations in the Rheb GTPase, one of the conserved TOR kinase regulators. We have found that unscheduled activation of Rheb resulted in the alteration of the proper localization of the pheromone receptor, Pra1, and thereby pheromone insensitivity. Since pheromone signaling triggers virulence in Ustilaginales, we believe that the Rheb-induced pheromone blindness was responsible for the associated lack of virulence. Strikingly, although these effects required the concourse of the Rsp5 ubiquitin ligase and the Art3 α-arrestin, the TOR kinase was not involved. Several eukaryotic organisms have shown that Rheb transmits environmental information through TOR-dependent and -independent pathways. Therefore, our results expand the range of signaling manners at which environmental conditions could impinge on the virulence of phytopathogenic fungi.

## Author summary

Fungal pathogens often use the adverse conditions found on plant surfaces (nutrient limitation and presence of some stresses) as triggers of the virulence program. Previous reports have shown that conditions or mutations resulting in unscheduled activation of the TORC1 pathway (which signals good environmental and internal signals to promote growth and proliferation in eukaryotic cells) disable the virulence program in phytopathogenic fungi. Using mutations that activate Rheb1, the GTPase responsible for TORC1 activation, we have addressed whether unscheduled TORC1 activity impairs the virulence of Ustilago maydis, the agent responsible for the corn smut disease. Strikingly, we have found that overactivation of Rheb1 resulted in an inhibition of virulence, but this effect was TORC1-independent. Several reports in other eukaryotic systems demonstrated non-

**Funding:** This work was supported by Ministerio de Ciencia e Innovación (MCIN) Grant PID2020-120242GB-I00 (MCIN/ AEI/10.13039/ 501100011033) to JPM. The funders had no role in study design, data collection and analysis, decision to publish, or preparation of the manuscript.

**Competing interests:** The authors have declared that no competing interests exist.

canonical roles of the Rheb regulatory axis. In this work, we added the control of virulence in U. maydis to this increasing list. We also tried to determine at which steps Rheb is controlling the virulence in U. maydis, and we have found that the ability to sense the presence of pheromone, the trigger of the virulence program in this fungus was affected.

## Introduction

For many phytopathogenic fungi, the activation of the virulence program is a consequence of the response to environmental cues found on plant surfaces, mainly chemical- and surface-dependent stimuli. This extracellular information interacts with the internal cell status (nutritional and energetic conditions) at various levels in a sort of "fungal brain," which integrates exogenous and endogenous signals to determine the coordinated cellular response. One of the more essential elements for appropriated signal integration in a eukaryotic cell is the TORC1 (Target Of Rapamycin Complex 1) pathway, widely conserved in fungi [1]. The central role of TORC1 is to integrate a large amount of cellular and environmental information, including growth and differentiation factors, cellular stress, and levels of nutrients and energy [2]. Integrating these stimuli enables a response that controls multiple facets of cellular metabolism. In conditions favoring cell growth (nutrients or growth factors rich conditions, absence of environmental stress), TORC1 is active promoting anabolic pathways (like, for instance, ribosome synthesis) and, at the same time, disabling catabolic pathways (like autophagy) [3]. Adverse conditions such as lack of nutrients or the presence of some stresses downregulate TORC1 activity, and therefore anabolic pathways are shut down, while catabolic pathways (including autophagy) are activated [4].

In phytopathogenic fungi, the current view is that conditions promoting high TORC1 activity disable the virulence program. This view was supported by studies showing that the repression of the virulence program by favorable nutritional conditions is released upon downregulating TORC1 activity by treatment with rapamycin in *Fusarium oxysporum* [5] or by using mutants in TORC1 effectors in *Fusarium graminearum* [6]. Extensive and elegant work done in the blast rice fungus, *Magnaporthe oryzae*, demonstrated that signaling mutations that mimic nutrient-favorable situations lead to an activated TORC1, which inhibits virulence. For instance, loss-of-function in the GATA factor Asd4 increased the intracellular pool of glutamine, which activated the TORC1 pathway and resulted in appressorium formation defects [7]. In the same way, glucose addition or mutations in *abl1*, encoding a glucose-controlled AMPK β subunit, also leads to appressorium defects via TORC1 activation [8]. Similarly, mutants in *whi2* (encoding a stress response regulator) from *M. oryzae* and the anthracnose disease-causing *Coletothrichum orbiculare* were avirulent. In both cases, the defects were linked to unscheduled activation of TORC1 in a still undefined manner, also resulting in defects in the appropriated development and function of the appressorium [9,10]. Studies in *M. oryzae* also uncover an intricate connection of TORC1 with other stimuli apart from nutrients, like surface-related cues, and how the integration of these signals resulted in decision-making under dynamic environmental conditions [11]. However, the molecular details of this connection and the targets of the TORC1 pathway responsible for virulence inhibition are still unknown.

Inspired by these reports, we have tried to address the consequences of TORC1 signaling activation during the induction of the virulence program in the corn smut fungus *Ustilago maydis*. The virulence program in *U. maydis* is linked to the sexual cycle [12,13]. Because of that, the pathogenic development in *U. maydis* depends on two independent sexual loci: the *a*-

locus, which encodes a pheromone-receptor system, and the *b*-locus, which encodes a pair of homeoproteins (bW and bE). On the plant surface, infection is initiated upon recognizing mating pheromone secreted by haploid cells of the opposite *a*-mating type. In each mating partner, this recognition activates a transcriptional program inducing the formation of long conjugation tubes, which grow toward each other and fuse at their tips resulting in a dikaryotic cell, the so-called infective filament. This specialized hypha explores the plant surface, and eventually, the tip swells to form the appressorium, which penetrates the cuticle. Significantly, the response to pheromone, which is the main trigger of the virulence program, is subdued to strict environmental control, and therefore, *a priori*, the involvement of the TORC1 pathway in this process is predicted.

The activation of the virulence program in *U. maydis* responds to several environmental conditions, from which nutrient availability is only one of them [14,15]. Because of this, instead of altering the fungus's nutritional perception, as described in the studies performed with *M. oryzae*, we decided to obtain mutations resulting in the direct activation of the TORC1 complex. TORC1 activity in eukaryotic cells, except for plants [16], is controlled by two distinct GTPase complexes (RAG and Rheb), which roles and essentiality depend on the studied organism. The first of these complexes comprises 4 GTPases in metazoan (RagA-D, called Ragulator) and 2 GTPases in fungi (Gtr1 and Gtr2). The role of this complex is to transmit information concerning amino acid availability in the cell. RAG GTPases have been described in metazoan and *Saccharomyces cerevisiae* as direct activators of TORC1, and gain-of-function mutants in this complex have been reported in these organisms [17]. However, recent reports in *Schizosaccharomyces pombe* suggested that the relationships between RAGs and TORC1 could be more complex [18,19] and that RAGs play a crucial role in attenuating more than activating TORC1 activity [20]. For these reasons, we focused on an alternative way to activate TORC1 by using the small GTPase Rheb, the second complex involved in TORC1 activation. Rheb conveys signals from environmental cues in metazoan, including growth factors, energy, and oxygen levels [21]. In fungi, the environmental stimuli transmitted by Rheb are unknown [22]. We describe in this work that Rheb activates TORC1 in *U. maydis*. We recreated in *U. maydis* the Rheb-activating mutations reported in other systems and demonstrated that these mutations' presence resulted in an environmental-independent TORC1 activation in *U. maydis*. Strikingly, we found that these Rheb-activating mutations inhibited virulence in *U. maydis*, but this effect was TORC1-independent. We traced the reasons for inhibition of the virulence to the early stages of the infective process, consisting of the ability of the fungus to detect the presence of compatible pheromones. Since Rheb can signal through TORC1-dependent and -independent pathways [23], our results expand the range of signaling manners at which environmental conditions could impinge on the virulence of phytopathogenic fungi.

## Results

### TORC1 is essential in *U. maydis*

The TORC1 pathway was not characterized in *U. maydis*, and we have made an initial effort to describe the main compounds of the TORC1 pathway in this fungus. Central to TORC1 is the TOR kinase. In contrast to other well-known systems like *S. cerevisiae* or *S. pombe*, in which there are two isoforms of TOR kinase encoded by distinct genes, in *U. maydis*, there is a single gene encoding this kinase (UMAG_03216, from here *tor1*, S1 Table) [1]. Our attempts to obtain a null mutant in *tor1* were unsuccessful, suggesting that the Tor1 function was essential in *U. maydis*. Therefore, we carried out two complementary manners to address the essentiality of Tor1.

We used a genetic approach consisting of the construction of a conditional allele of *tor1* by the exchange of its native promoter with the regulatable promoter P*nar1* (which is controlled by the nitrogen source, being active in the presence of nitrate and inactive in the presence of ammonium and amino acids [24] (S1A Fig)). Cells carrying the *tor1*[nar1] allele incubated in restrictive conditions (rich medium, YPD) showed decreased levels of *tor1* mRNA (although some level of transcript remained, most likely because of the leakiness of the conditional promoter, S1B Fig), which correlated with impaired ability to grow in solid medium (Fig 1A). We also noted that the morphology of conditional cells incubated in restrictive conditions changed. Cells were swollen in the medial region (Fig 1C), most likely as a consequence of vacuole enlargement (S2 Fig), a typical response to TORC1 inhibition in other fungi [25].

We also used a chemical approach using two well-described chemical inhibitors acting on TOR kinase: Torin1 and rapamycin [26]. Torin1 is an ATP-analog that directly targets the TOR kinase [27]. Rapamycin acts indirectly, binding to the proline isomerase FKBP12 and forming a complex that interacts explicitly with TORC1, inhibiting its activity [28,29]. The ability of rapamycin to inhibit growth is variable in distinct organisms, likely related to the drug's affinity to the FKBP12 protein [30]. Although both drugs affected the growth of *U. maydis* cells in a solid medium, the effects of torin1 were more drastic than those caused by rapamycin (Fig 1B). These differences were also observed in liquid cultures (S3 Fig). As in the conditional *tor1*[nar1], we observed a change in the cells' morphology, more severe in the case of torin1 treatment, compatible with an enlargement of vacuoles (Figs 1D and S2). Altogether, these data supported the conclusion that Tor1 seems essential in *U. maydis*.

The TOR kinase in most eukaryotes is part of two different complexes, TORC1 and TORC2. It was unclear in *U. maydis* whether the essentiality of Tor1 was attributable to TORC1, TORC2, or both complexes. TORC1 and TORC2 differ in the signals they perceive, their regulated processes, and the accessory proteins that make them up. TORC1 contains the RAPTOR (Regulatory Associated Protein of TOR) protein, while TORC2 contains two specific proteins called RICTOR (Rapamycin Insensitive Companion of TOR) and SIN1 (Stress-activated protein kinase INteracting protein 1) [31]. In the distinct TOR complexes, these associated subunits act as scaffold proteins, which provide specificity of action to each complex, favoring the interaction with their respective effectors. Besides, both complexes shared a regulatory subunit called LST8. Analysis of the *U. maydis* genome indicated the presence of predicted genes encoding all these proteins, which we have named *rpt1* (UMAG_00801), *rct1* (UMAG_06215), *sin1* (UMAG_00947), and *lst8* (UMAG_03059) (S1 Table). We tried to obtain loss-of-function alleles in the genes encoding the respective scaffolding proteins. We successfully deleted *rct1* and *sin1* (TORC2) but not *rpt1* (TORC1). Cells lacking TORC2 were slightly affected in growth (assessed as colony size in solid medium) and increased cell length (S4 Fig). Respecting TORC1, we constructed cells carrying a *rpt1*[nar1] conditional allele (S1C and S1D Fig), which, when grown in restrictive conditions, phenocopied the defects observed in *tor1*[nar1] (Figs 1A and 1C and S2). These results indicated that TORC1, but not TORC2, was essential in *U. maydis*.

TORC1 controls multiple facets of cellular metabolism. Two of them, ribosomal gene transcription and autophagy, can be easily used as indirect readouts of TORC1 activity. Autophagy, which TORC1 negatively regulates, can be monitored by using a GFP-Atg8 fusion. The appearance of free GFP on Western blots is used to measure autophagic flux (the GFP half is not sensitive to vacuole proteases) [32]. Besides, ribosomal transcription, which TORC1 positively controls, can be analyzed by qRT-PCR of two different *U. maydis* ribosomal genes, *rpl43* and *s21b*. We analyzed autophagy and ribosomal mRNA levels in cells carrying t*or1*[nar1] or *rpt1*[nar1] alleles and in cells treated with torin1 and rapamycin. The results were coherent with

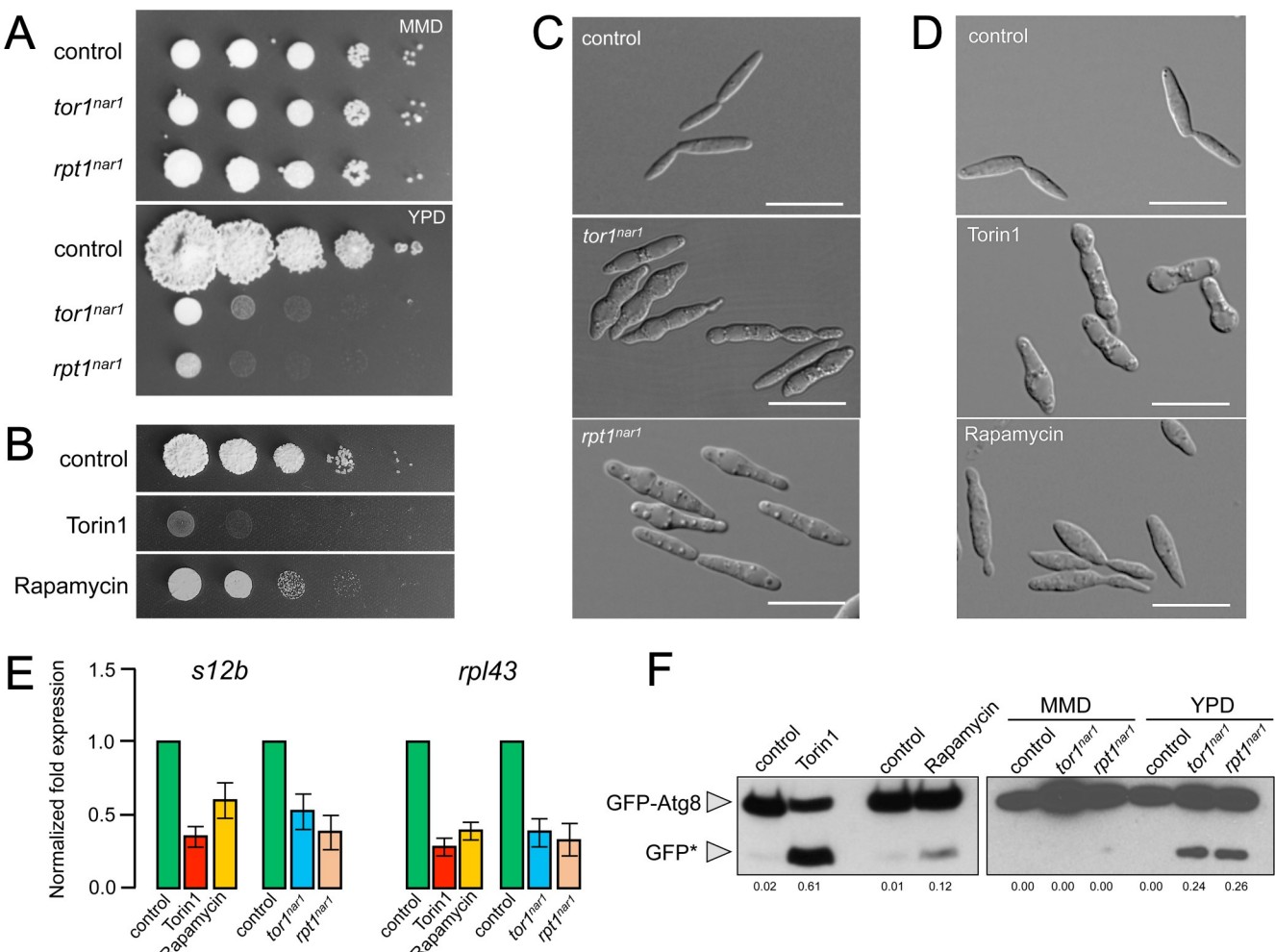

**Fig 1. TORC1 is essential in *U. maydis*.** (A) Conditional alleles for the TORC1 components *tor1* and *rpt1*. Serial tenfold dilutions of cultures from strains carrying the indicated conditional alleles (control: FB1) were spotted in solid nitrate minimal medium (MMD, permissive conditions for $P_{nar1}$ expression) or solid yeast-peptone medium (YPD, restrictive conditions for $P_{nar1}$ expression). Plates were incubated for 3 days at 28°C. (B) Chemical inhibition of TORC1. Serial tenfold dilutions of cultures from FB1 cells spotted in YPD were amended with 10 μM Torin1 and 1μg/ml rapamycin. Plates were incubated for 3 days at 28°C. Control plates included DMSO (1% final) as solvent. (C) Representative images of liquid cultures of strains carrying the indicated conditional alleles (control: FB1), incubated in YPD for 8 hours. Bar: 20 μm. (D) Representative images of liquid cultures of FB1 cells grown in YPD for 8 hours in the presence of 10 μM Torin1 and 1μg/ml rapamycin. Bar: 20 μm. (E) TORC1 is required for ribosomal gene expression. qRT-PCR for the indicated ribosomal genes from FB1 cells incubated for 8 hours in YPD in the presence of 10 μM Torin1 and 1μg/ml rapamycin (left block of columns) or strains carrying the indicated conditional alleles (control: FB1, right block of columns) incubated for 8 hours in YPD. Each column represents the mean value of three independent biological replicates. Values are referred to the expression of each gene in the control strain. Error bars represent the SD. (F) TORC1 inhibits autophagy. Western blot of extracts from strains cells carrying an endogenous GFP-Atg8 allele. Cells were incubated for 8 hours in YPD liquid medium for chemical inhibition in the presence of 10 μM Torin1 and 1μg/ml rapamycin. Conditional alleles were incubated for 8 hours in a minimal nitrate medium (MMD, permissive condition) or YPD (restrictive condition). A similar amount of protein extracts was separated by SDS-PAGE. Immunoblots were incubated with an antibody against GFP. The ratio between the signal corresponding to free GFP (GFP*) and GFP-Atg8 (which indicated the autophagic flux) for each lane is shown at the bottom of the gel (average from 3 independent westerns).

the role of TORC1 in the regulation of central metabolism described in other systems: TORC1 seemed to promote ribosomal synthesis and repress autophagy in *U. maydis* (Fig 1E and 1F).

## TORC1 feeds two distinct AGC kinases in *U. maydis*

We were interested in finding manners to keep active TORC1 independent of environmental conditions. In the previous section, we used the transcription level of ribosomal gene mRNAs

and autophagy as readouts for TORC1 activity. However, in many organisms, both readouts are subject to additional layers of regulation by other regulatory cascades like Protein Kinase A or the Gcn2 pathways [33,34]. Therefore, we seek more specific readouts of TORC1 activity.

TORC1 signals through many effectors, mostly kinases activated or repressed by the Tor kinase-mediated phosphorylation [35]. One of the more conserved effectors is S6K (S6 Kinase), which upon activation by TORC1, phosphorylates the Rps6 ribosomal protein [36,37]. The degree of S6K-dependent Rps6 phosphorylation is a widely-used reporter of TORC1 activity because of the conservation of phosphorylated residues in Rps6 (corresponding to Ser235 and Ser236 in *Homo sapiens* Rps6) and the availability of commercial antibodies against these phosphorylated residues [38]. Unfortunately, although this phosphorylation site is highly conserved in fungi, it was not in *U. maydis* (S5 Fig). Therefore, we sought alternative ways to address TORC1 activity in *U. maydis*.

In fungi, several AGC kinases were located downstream of TORC1 or TORC2 [35]. A BLASTp search of the *U. maydis* genome revealed two putative AGC kinases predicted by sequence to be downstream of the TOR pathway (S6 Fig). Aga1 was already characterized as a kinase required for virulence in *U. maydis* [39]. Aga1 is similar to the *S. pombe* Gad8 and *S. cerevisiae* Ypk1 and Ypk2, all TORC2 effectors [36,40,41]. The other kinase, so far uncharacterized (UMAG_00602, from here Sch9), was similar to *S. pombe* Sck1 and Sck2 and *S. cerevisiae* Sch9, which were located downstream of TORC1 [42,43]. Neither Aga1 nor Sch9 was essential for *U. maydis*, although single mutations affected the growth in solid medium, and double mutants showed an additive effect (Fig 2A). We tried to tag the C-terminus end of both proteins with a triple HA epitope. While Aga1-3HA was fully functional [39], the tagging of Sch9 (either N- or C-terminal end) resulted in an inactive protein.

We analyzed Aga1-3HA immunoprecipitates in Phostag-containing SDS gels, and we observed that upon treatment with torin1, rapamycin, or λ-phosphatase, the electrophoretic mobility of Aga1-3HA increased (Figs 2B and S7A), suggesting Tor1-dependent phosphorylation. To address which Tor1-containing complex was responsible for this electrophoretic shift, we analyzed the electrophoretic mobility of Aga1-3HA immunoprecipitates from mutants lacking either Sin1 (TORC2) or down-regulating Rpt1 (TORC1) (Fig 2B). Unexpectedly, our results suggested that in *U. maydis*, Aga1 was under the control of TORC1, which contrasts with the phylogenetic analysis that predicted Aga1 to be part of TORC2 and Sch9 to be part of TORC1 (S6 Fig).

We were curious about the distinct wiring between TOR complexes and AGC kinases in *U. maydis*, and we wondered whether Sch9 was also downstream of TORC1. Since we could not tag Sch9 to analyze possible Tor1-dependent electrophoretic mobility changes, we used an indirect approach to address the activity of Sch9, respecting the distinct TOR complexes. In other systems, the transcriptional regulator Maf1 was phosphorylated by Sch9 [44]. We tagged with triple HA the Maf1 homolog found in the *U. maydis* genome (UMAG_11056, S8A Fig). Using Phostag-containing SDS gels, we observed that torin1, rapamycin, or λ-phosphatase treatment increased the electrophoretic mobility of Maf1-3HA immunoprecipitates (S8B Fig). The mobility change was also observed when *sch9* was deleted, while the deletion of *aga1* showed no effect, supporting Maf1 as a target of Sch9 (Fig 2C). We also analyzed whether the change of mobility of Maf1-3HA immunoprecipitates was dependent on Rpt1 or Sin1, and we found it to be TORC1 dependent (Fig 2C). Therefore, we propose that in *U. maydis*, Aga1 and Sch9, were located downstream of TORC1.

Studies in other fungal systems showed that TORC1 activity responds to growth medium conditions depending on the nitrogen source, being more active in media using amino acids as a nitrogen source and decreasing its activity in less-favorable nitrogen sources [45]. Using the changes of electrophoretic mobility of Aga1-3HA and Maf1-3HA immunoprecipitates, we analyzed whether this was also the case for *U. maydis*. We observed that in cells grown in a

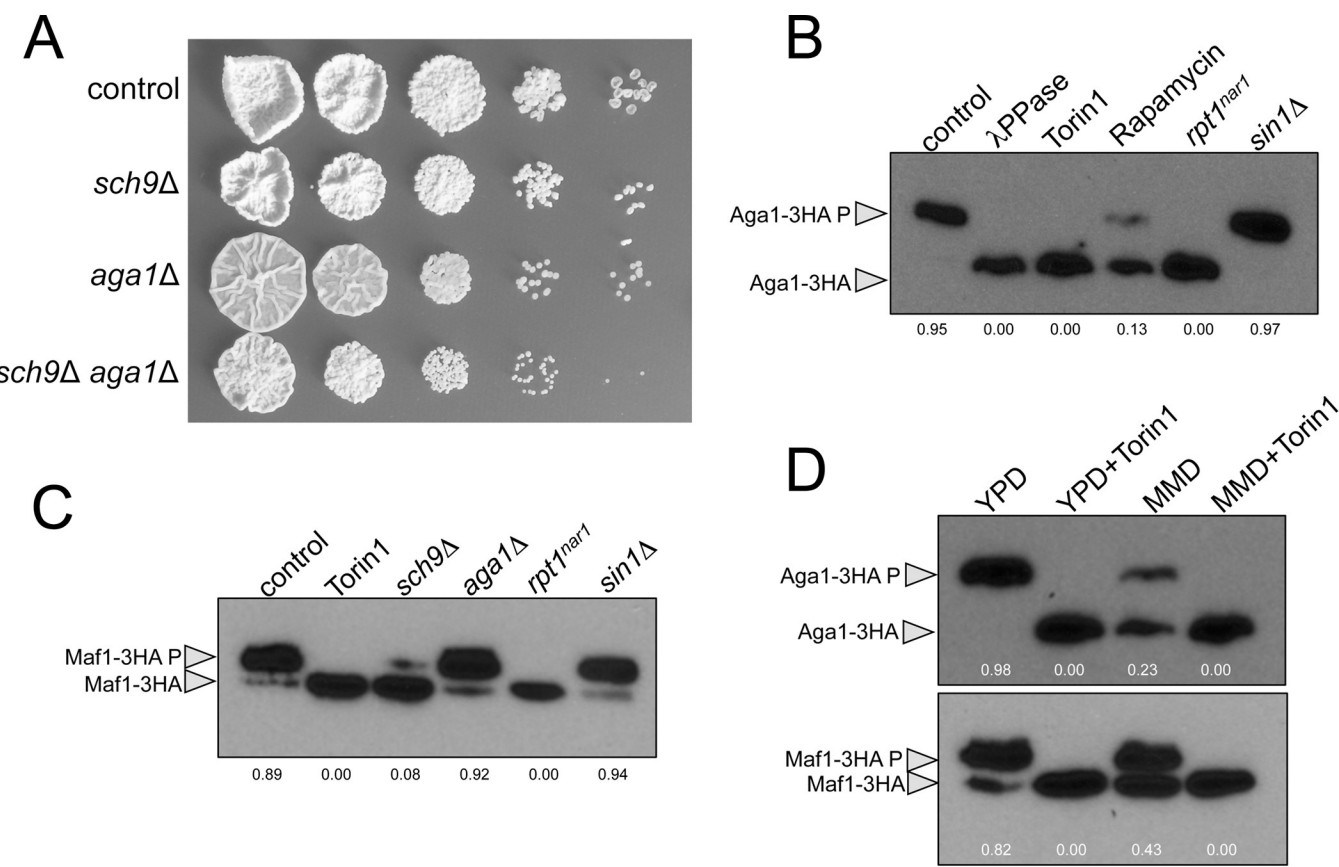

**Fig 2. TORC1 signals through two AGC kinases in *U. maydis*.** (A) Loss-of-function alleles for AGC kinases. Serial tenfold dilution of cultures from strains carrying the indicated deletion alleles (control: FB1), spotted in solid YPD medium. Plates were incubated for 3 days at 28°C. (B) TORC1-dependent phosphorylation of Aga1. Anti-HA immunoprecipitates of cell extracts from cultures of cells carrying an endogenous Aga1 HA-tagged allele grown for 8 hours in YPD were incubated at 30°C for 20 min in the absence (control) or presence (λ PPase) of lambda protein phosphatase. They were then separated in Phos-tag gels and subjected to immunoblot analysis with anti-HA. The gel also showed anti-HA immunoprecipitates of cell extracts from cultures of control cells grown in YPD amended with 10 μM Torin1 and 1μg/ml rapamycin and cultures from the indicated mutant strains, also grown 8 hours in YPD. Numbers below each gel lane indicated the ratio between the phosphorylated and unphosphorylated bands and results from the average of three independent western assays. (C) TORC1-dependent phosphorylation of Maf1. Anti-HA immunoprecipitates of cell extracts from cultures of cells carrying an endogenous Maf1 HA-tagged allele and the indicated mutations grew for 8 hours were separated in Phos-tag gels and subjected to immunoblot analysis with anti-HA. Control cultures were also incubated with 10 μM Torin1. Numbers below each gel lane indicated the ratio between the phosphorylated and unphosphorylated bands and results from the average of three independent western assays. (D) TORC1-dependent phosphorylation and nutritional conditions. Western blot anti-HA immunoprecipitates from cell extracts obtained from cultures of cells carrying an endogenous Maf1 or Aga1 HA-tagged allele incubated for 8 hours in YPD or nitrate minimal medium (MMD) with or without the addition of 10 μM Torin1. Numbers below each gel lane indicated the ratio between the phosphorylated and unphosphorylated bands and results from the average of three independent western assays.

minimal nitrate medium, where the sole nitrogen source was nitrate, a higher proportion of Aga1-3HA and Maf1-3HA showed faster electrophoretic mobility than the mobility from cell extracts obtained from cells grown in yeast-peptone medium (Fig 2D).

In summary, our results support the notion that in *U. maydis*, TORC1 signals through two AGC kinases, Aga1 and Sch9, and that the changes in electrophoretic mobility of Aga1-3HA and Maf1-3HA immunoprecipitates can be used as readouts of TORC1 activity. We also provided evidence that TORC1 activity responds to nutritional conditions.

## Rheb GTPase regulates TORC1 in *U. maydis*

The Rheb GTPase has been involved in controlling TORC1. We have found a homolog of Rhb1 in the *U. maydis* genome (S1 Table), and our attempts to delete this gene were

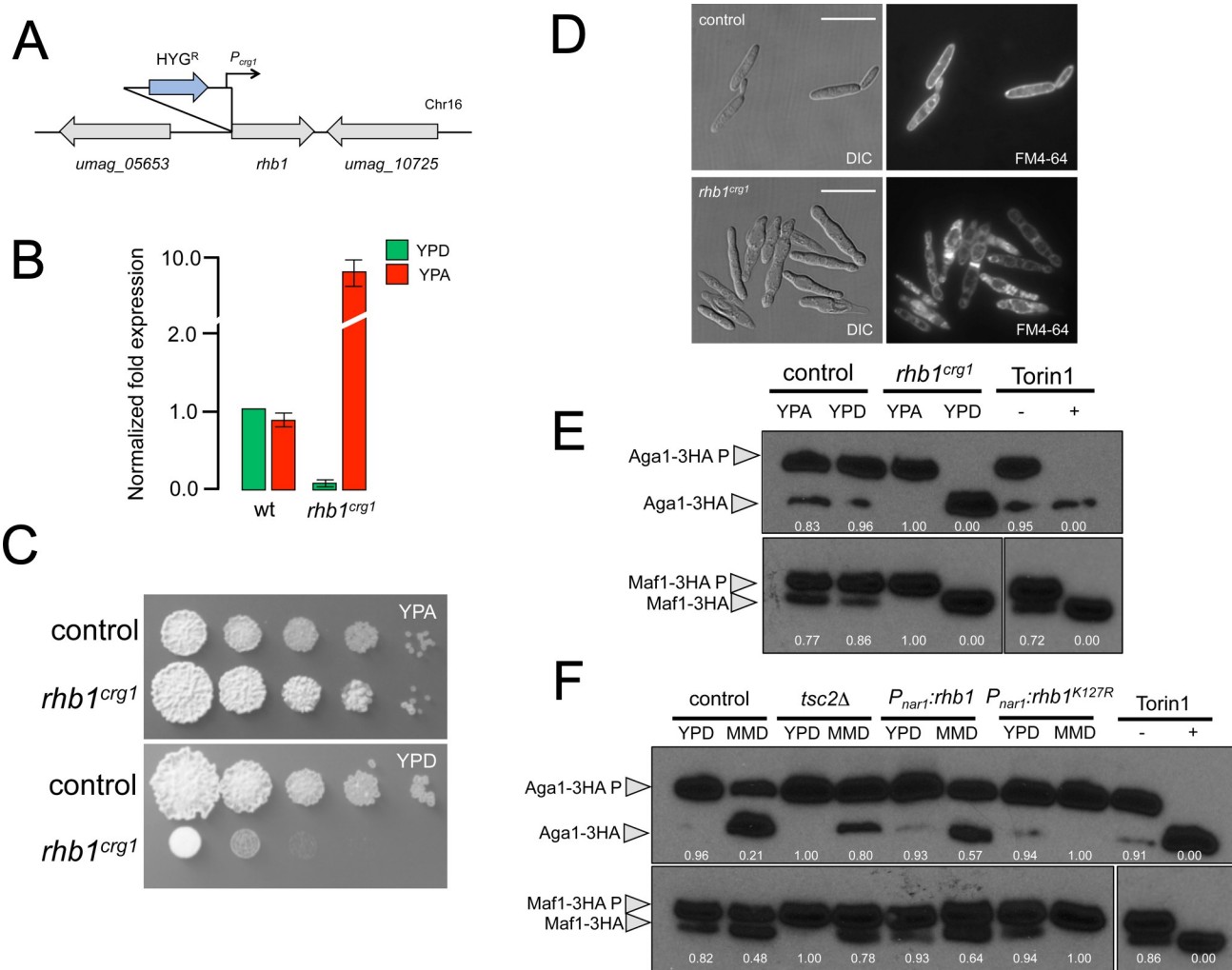

**Fig 3. Rheb GTPase is required for TORC1 activation in *U. maydis*.** (A) Scheme of *rhb1^crg1* conditional allele. (B) qRT-PCR of *rhb1* mRNA levels from control and *rhb1^crg1* conditional cells, incubated for 8 hours in YPD (glucose, repressive conditions) or YPA (arabinose, permissive conditions). Values are referred to the expression of *rhb1* in FB1 (control strain) grown in YPD. Each column represents the mean value of three independent biological replicates. Error bars represent the SD. (C) Serial tenfold dilutions of cultures from strains carrying the *rhb1^crg1* conditional allele (control: FB1), spotted in solid yeast peptone arabinose medium (YPA, permissive conditions for P_crg1 expression) or solid yeast-peptone glucose medium (YPD, restrictive conditions for P_crg1 expression). Plates were incubated for 3 days at 28°C. (D) Representative images of liquid cultures of cells carrying the *rhb1^crg1* conditional allele indicated conditional alleles (control: FB1), incubated in YPD for 8 hours. Fluorescent images showed FM4-64 staining. Bar: 20 μm. (E) Quantification of TORC1 activity in *rhb1^crg1* cells using the electrophoretic mobility change of Aga1-3HA and Maf1-3HA. Indicated strains were grown for 8 hours in YPD or YPA, and anti-HA immunoprecipitates from cell extracts were separated in Phos-tag gels and subjected to immunoblot analysis with anti-HA. Control cultures incubated with or without 10 μM Torin1 were included as a control for mobility changes. Numbers below each gel lane indicated the ratio between the phosphorylated and unphosphorylated bands and results from the average of three independent western assays. (F) Quantification of TORC1 activity in strains carrying mutations that upregulate Rheb, using the electrophoretic mobility change of Aga1-3HA and Maf1-3HA. Indicated strains were grown for 8 hours in YPD (control conditions) and minimal medium amended with nitrate (MMD), and anti-HA immunoprecipitates from cell extracts were separated in Phos-tag gels and subjected to immunoblot analysis with anti-HA. Control cultures incubated with or without 10 μM Torin1 were included as a control for mobility changes. Numbers below each gel lane indicated the ratio between the phosphorylated and unphosphorylated bands and results from the average of three independent western assays.

unsuccessful. However, we obtained a conditional allele exchanging its promoter with the *crg1* promoter controlled by the carbon source (repressed by the presence of glucose and activated by arabinose) [24] (Fig 3A and 3B). In repressive conditions (YPD), cells carrying the conditional allele were impaired in growth (Fig 3C) and phenocopied the morphological defects observed when TORC1 was down-regulated (Fig 3D). These results were consistent with the

Rheb proposed role of activating TORC1. To further support Rhb1 as a TORC1 regulator in *U. maydis*, we analyzed the electrophoretic mobility of Aga1-3HA and Maf1-3HA immunoprecipitates in *rhb1^crg1* conditional cells grown in permissive and restrictive conditions (Fig 3E). We found that Rhb1 was required to maintain low electrophoretic mobility of both Aga1-3HA and Maf1-3HA, supporting the notion that *U. maydis* Rhb1 was required to activate TORC1.

Rheb, in its GTP-bound form, activates the TORC1 while it is inactive in its GDP-bound form. The GTP/GDP status of Rheb is controlled by the TSC complex, composed of Tsc2 (Tuberin, the GTPase-activating protein) and Tsc1 (Hamartin, required to stabilize Tsc2) [46]. We have also found putative homologs of Tsc1 and Tsc2 proteins in the *U. maydis* genome (S1 Table).

We reasoned that altering Rheb activity in *U. maydis* would allow us to over-activate TORC1 at will. For that, we attempted two strategies. We made a loss-of-function mutant of *tsc2*, which was viable. We also constructed a hyperactive allele of Rhb1 carrying a point mutation (K127R), which in *S. pombe* confers constitutive activity to Rhb1 [47]. The *rhb1^K127R* allele was cloned under the control of *the nar1* promoter (and therefore, its expression induced in nitrate-amended medium) and integrated into the *U. maydis* genome at the *ip* locus, a frequent site to insert ectopic copies of genes in *U. maydis* [48,49] (S9 Fig). We analyzed the effects in the TORC1 activity of the *tsc2* removal or the expression of an ectopic *rhb1^K127R* allele. Using the electrophoretic mobility of Aga1-3HA and Maf1-3HA immunoprecipitates as readouts, we found that in these mutants, the activity of TORC1 remained high even when cells were transferred to a minimal nitrate medium, a condition that in control cells resulted in low TORC1 activity (Fig 3F). Using these mutants, we were able to maintain the high activity of the TORC1 pathway even under poor nutritional conditions.

## Overactivation of TORC1 affects the ability of cells to grow in a minimal nitrate medium by inducing the degradation of the nitrate permease

During the previous experiments, we observed that the growth in a liquid medium of cells carrying the insertion of the *rhb1^K127R* allele was severely impaired in conditions inducing the expression of that ectopic allele (minimal medium amended with nitrate). We analyzed this observation in a solid medium and found that strains carrying the ectopic *rhb1^K127R* allele under the *nar1* promoter could not produce colonies in minimal nitrate medium plates. We also observed that cells lacking Tsc2 were likewise affected in growth in a minimal nitrate medium, although to a lesser extent (S10A Fig). Significantly, the growth impairment was suppressed when rapamycin -which in *U. maydis* is a mild inhibitor of TORC1- was added to the medium, strongly suggesting that the observed defect in growth was a consequence of the maintenance of high TORC1 activity (S10A Fig). Since the overactivation of TORC1 could be affecting the growth in a medium with a poor nitrogen source (like the nitrate), we sought to disentangle the expression of the ectopic *rhb1^K127R* allele from the nitrogen source. For that, we expressed an ectopic copy of the *rhb1^K127R* allele under the control of the P*crg1* promoter, which is regulated by the carbon source (glucose off, arabinose on), and it is not affected by the nitrogen source (S10B Fig). We found that cells incubated in a rich medium amended with arabinose (YPA, conditions enabling the expression of the ectopic *rhb1^K127R* allele) displayed some impairment in growth compared to glucose condition (YPD, conditions resulting in repression of the ectopic *rhb1^K127R* allele). However, this effect was not as dramatic as the consequences of the expression of the ectopic *rhb1^K127R* allele in the minimal nitrate medium amended with arabinose, where no growth was observed (S10C Fig). We think that these results can be explained by the view that keeping internal signaling (promoted by the forced high TORC1 activity) not compatible with environmental conditions (low quality of nitrogen source) was detrimental to cells.

The main aim of our work was to address the effect of forced TORC1 activation during the infection of plants by *U. maydis*, and for that, we were planning to use the Rheb-activating mutations. However, we considered it necessary to understand the reasons for the observed TORC1-associated growth defect when using these mutations because it could be affecting virulence by indirect causes (like inhibiting the growth of the cells). In budding and fission yeast, TORC1 adjust the growth in distinct nitrogen sources by establishing a hierarchy in the respective transporters to be degraded, inactivated, or maintained active at the plasma membrane (PM) [50–52]. We reasoned that one possible explanation for the observed growth defect was the inhibition of nitrate uptake, which in fungi depended on nitrate permeases located at PM [53] in conditions of high TORC1 activity. This inhibition would result in nitrogen starvation conditions, despite the presence of nitrate in the growth medium. To address this possibility, we analyzed the proposed nitrate permease from *U. maydis* (UMAG_11105, *nrt1* from here, see S11 Fig for characterization of this permease) [54,55]. In control cells, a GFP-tagged allele of Nrt1 was located at PM. We have also observed a substantial proportion of the GFP fluorescence associated with vacuoles compatible with the proposed mechanism of permease recycling using the endocytic/vacuole pathway observed in other fungi [56]. Strikingly, the ectopic *rhb1^{K127R}* allele (upon incubation in a minimal nitrate medium) resulted in the accumulation of the fluorescent signal at vacuoles, but no signal was found at PM. More importantly, the treatment of these cells with rapamycin recovered the Nrt1-GFP signal at PM (Fig 4A and 4B). Disabling the ability of rapamycin to inhibit TORC1 by deletion of the gene encoding the FKBP12 protein resulted in no recovery of the Nrt1-GFP signal at PM upon rapamycin treatment (S12 Fig).

We also analyzed by Western blot using anti-GFP antibodies the levels of the Nrt1-GFP protein, and we found that the signal corresponding to the full-length Nrt1-GFP was absent in the cells growing in nitrate and expressing *rhb1^{K127R}* allele but that this signal is recovered when rapamycin was added to these cells (Fig 4C). We believe that in cells growing in nitrate but having a forced high TORC1 activity, the nitrate permease was submitted to vacuolar degradation. We based this conclusion on two observations. First, we have discarded the observed down-regulation of nitrate permease levels at the transcriptional level. We analyzed by qRT-PCR the levels of *nrt1* mRNA, and we found that the presence of nitrate induced the expression of *nrt1* and that the constitutive TORC1 activity did not impede its induction (Fig 4D). Secondly, we observed in Western blot from cells growing in nitrate and expressing *rhb1^{K127R}* allele that, although no Nrt1-GFP band was apparent, it was possible to observe a clear signal with the expected size of free GFP, suggesting that the GFP-Nrt1 fusion protein pool was hydrolyzed in the vacuole (the GFP half is not sensitive to the proteases resident in vacuoles) (Fig 4C).

Studies performed in *S. cerevisiae* and *S. pombe* indicated that high TORC1 activity promoted the ubiquitylation of distinct amino acids permeases or their corresponding α-arrestins by the HECT-type E3 ubiquitin ligase Rsp5/Pub1, resulting in either on the vacuolar degradation or the intramembrane (mostly Golgi) accumulation of the targeted permeases [57–61]. We have found a putative homolog of Rsp5/Pub1 in the *U. maydis* genome (UMAG_00663). Encouragingly, cells expressing the ectopic *rhb1^{K127R}* allele and lacking Rsp5 were able to grow in minimal nitrate medium (Fig 4E) and this ability correlated with the stabilization of the nitrate permease, assessed by Western Blot (Fig 4C) as well as by its presence at PM (Fig 4A and 4B). In other fungi, Rsp5/Pub1 seems to work in concert with α-arrestins, which are at the same time effectors and regulatory subunits, providing specificity to the E3 ubiquitin ligase [62]. We have predicted four putative α-arrestins in the genome of *U. maydis* (S13A Fig). We obtained loss-of-function mutants in three of them, although no one of these appears to be

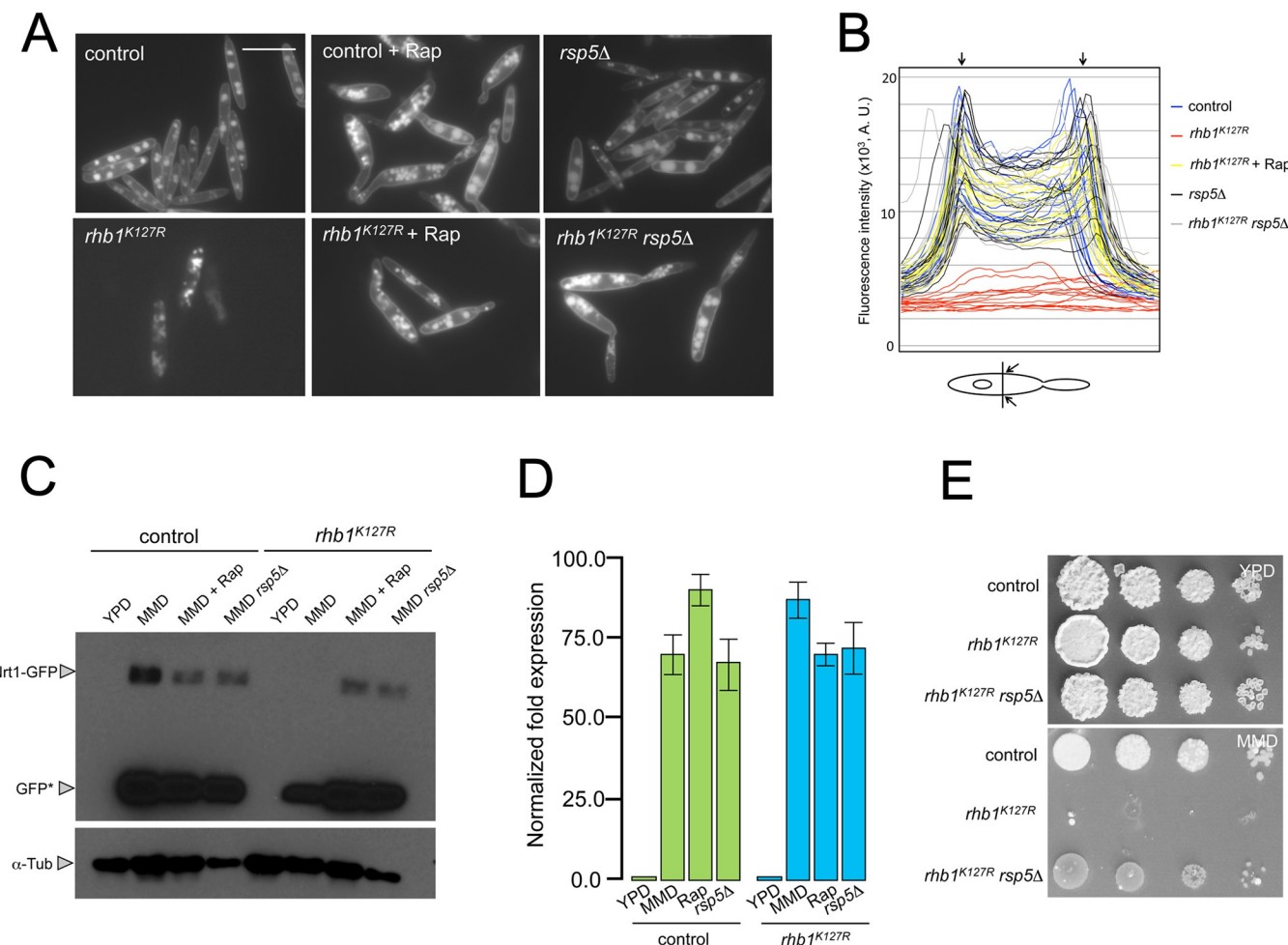

**Fig 4. Forced activation of Rheb downregulates the nitrate permease.** (A) Nitrate permease is not localized in cells expressing the *rhb1^{K127R}* allele at the plasma membrane. Fluorescence images of cultures from the indicated strains carrying an *nrt1-GFP* endogenous allele. Cells were grown for 8 hours in a minimal nitrate medium. Rapamycin (Rap) was added at 1μg/ml. All images were obtained using the same exposition time. Bar: 15 μm. (B) Line-scan analysis of Nrt1-GFP signal intensities of cells from (A). At least 10 cells from each strain and condition were scanned through its medial zone. Arrows indicate the location corresponding to the plasma membrane. A. U., arbitrary units. (C) Nrt1 is degraded in cells expressing the *rhb1^{K127R}* allele in a TORC1-dependent manner. Western blot analysis of cell extracts with anti-GFP antibodies from the indicated strains carrying an *nrt1*-GFP allele and expressing or not the *rhb1^{K127R}* allele, grown in different media for 8 hours. Rapamycin (Rap) was added at 1μg/ml. Levels of Tub1 were used as a loading control (bottom blot). (D) qRT-PCR for *nrt1* from control (FB1 P_{crg1}:*rhb1*) and cells expressing the *rhb1^{K127R}* allele incubated for 8 hours in the indicated conditions (Rap: minimal nitrate medium amended with 1μg/ml Rapamycin; *rsp5Δ* cells were incubated in minimal nitrate medium). Relative mRNA levels were referred to as the expression of *tub1*. Each column represents the mean value of three independent biological replicates. Error bars represent the SD. (E) Loss-of-function of Rsp5 suppresses the growth defects associated with the expression of the *rhb1^{K127R}* allele in a minimal nitrate medium. Serial tenfold dilutions of cultures from the indicated strains (control: FB1 P_{crg1}:*rhb1*) were spotted in solid yeast peptone or minimal nitrate medium. YPD and MMD plates were incubated for 3 and 4 days at 28˚C.

involved in the above-described down-regulation of Nrt1 in the presence of high TORC1 activity (S13B Fig).

In summary, our results indicated that the growth defect associated with Rheb-activating mutations was a consequence of misleading signal integration by the cell, resulting in the inability to grow in a medium containing nitrate as the only nitrogen source, most likely because of the unappropriated vacuolar degradation of the nitrate permease. Furthermore, we have found that this defect in growth for strains carrying Rheb-activating mutations was specific for nitrate as a nitrogen source, and it was not found when cells were growing in a minimal medium amended with ammonium (see below).

## The presence of Rheb-activating mutations inhibits the ability of *U. maydis* to infect plants

Nitrate is an unlikely nitrogen source during the plant infection, and therefore we believe that the above-described growth defect associated with high TORC1 activity would not affect *U. maydis* cells during the infection process. We sought to bypass the dependence on nitrate for expression of the *rhb1^{K127R}* allele, and therefore we designed a strategy to introduce the K127R mutation into the endogenous *rhb1* locus using the CRISPR/Cas9-based technology (see Materials and Methods). We introduced the Rheb-activating mutations into sexual compatible (FB1 and FB2, carrying *a1 b1* and *a2 b2* mating-type respectively) and solopathogenic (SG200, harboring a compatible combination of *a* and *b* mating types) strains and addressed their ability to produce colonies in different growth media. Importantly, we found that they grew like control cells in rich (YPD) and minimal medium amended with ammonium and showed a growth reduction in minimal medium amended with nitrate, most likely because of the down-regulation of nitrate transporter (S14A Fig). We also addressed TORC1 activity (using as a readout the electrophoretic mobility of Maf1-3HA) in a minimal medium amended with ammonium (a medium in which the growth of the cells seemed not to be compromised), and we found that in these conditions, the strains carrying the *rhb1^{K127R}* and *tsc2Δ* alleles showed higher TORC1 activity than control cells (S14B Fig).

We could maintain a high TORC1 activity by using the Rheb-activating mutations, so we asked whether this forced TORC1 activity would affect the virulence of *U. maydis* cells. We cultured the distinct strains in a minimal medium amended with ammonium and infected corn plants with compatible cell mixtures and solopathogenic cells from control and the respective mutant strains. Encouragingly, we found that strains carrying Rheb-activating mutations were avirulent (Fig 5). With these results in hand, we aimed to address how the presence of Rheb-activating mutations interferes with the virulence in *U. maydis*.

## The presence of Rheb-activating mutations leads to TORC1-independent insensitivity to pheromone signaling

Corn infection by *U. maydis* cells is strictly dependent on the infective filament. An easy way to address this structure's formation is to spot the cell mixtures on charcoal-containing plates. On this substrate, the formation of infective filaments upon mating resulted in white, fuzzy colonies (*fuz* phenotype) [63]. We analyzed the *fuz* phenotype in cell mixtures of control and mutant strains as well as in solopathogenic derived strains (SG200 is independent of mating partner) [64], and we found that no infective filaments were produced when the Rheb-activating mutations were present (Fig 6A). This result suggested that the more likely reason for the lack of virulence from strains carrying the Rheb-activating mutations was their inability to produce the infective filament.

One appealing explanation for the absence of infective filament formation was that the transcriptional program associated with the sexual cycle (which includes the genes from the *a*- and *b*-loci) was not induced in the presence of the Rheb-activating mutations. This transcriptional program is triggered by the presence of compatible pheromones [65]. We treated *a1* mating-type control and mutant strains grown in minimal ammonium medium with synthetic a2 pheromone to address this possibility. We analyzed the transcriptional levels of *mfa1* and *pra1* (*a1* locus), *bE1* (*b1* locus), and *prf1* (the transcriptional activator of mating-type genes) by qRT-PCR (Fig 6B). We also analyzed the formation of conjugation tubes as a readout of the pheromone response (Fig 6C). Strikingly, we found that in strains carrying the *rhb1^{K127R}* and *tsc2Δ* alleles, the cells were unresponsive to compatible pheromone: they do not produce conjugation tubes. They do not increase the mRNA levels of the mating-type genes.

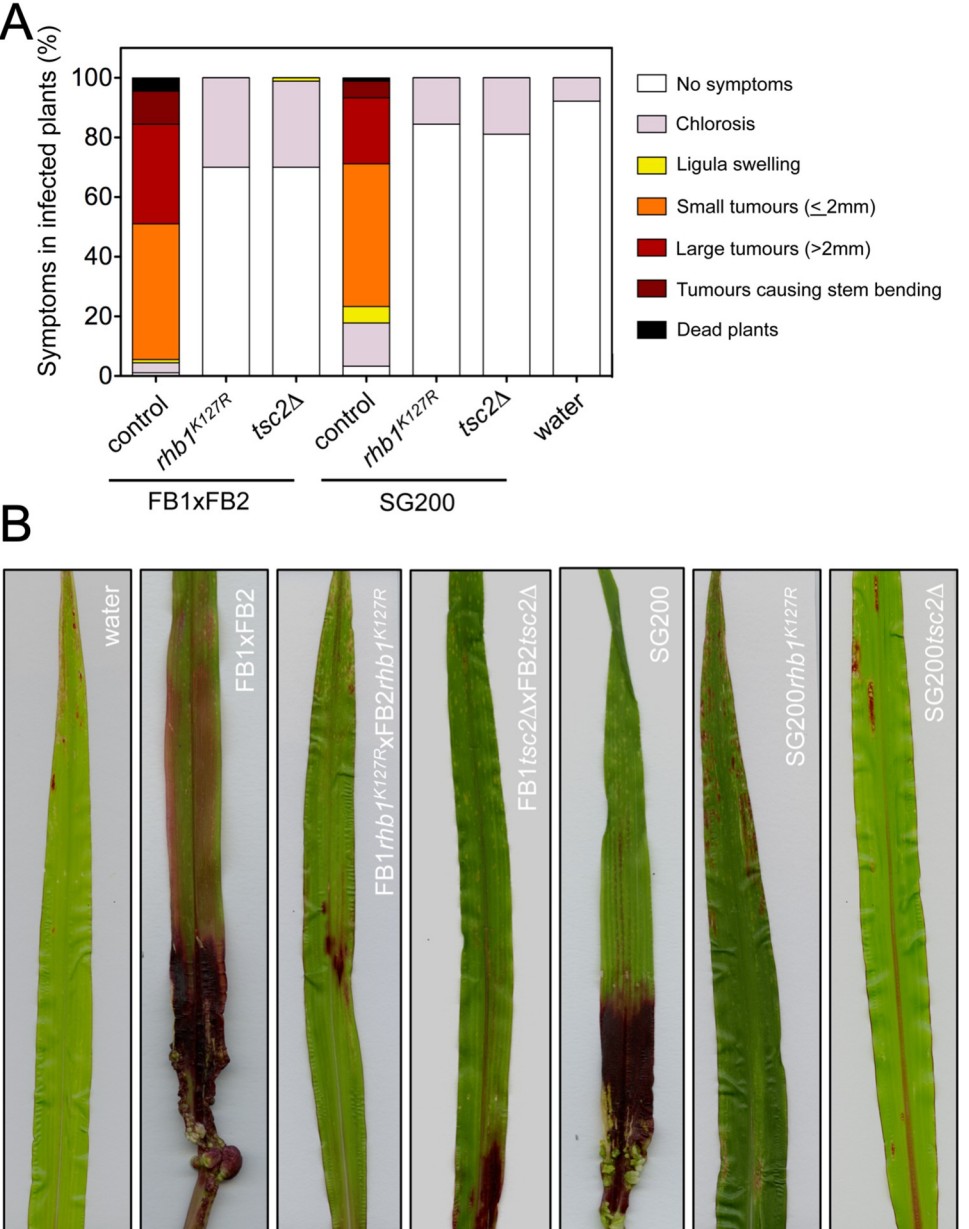

**Fig 5. Forced activation of Rheb resulted in avirulence.** (A) Graph showing disease symptoms caused by wild-type and mutant strains cross. The symptoms were scored 14 days after infection. Three independent experiments were carried out, and the average values are expressed as a percentage of the total number of infected plants (n: 30 plants in each experiment). (B) Representative leaves 14 days after infection with the indicated strains.

We wondered whether down-regulation of TORC1 upon rapamycin treatment activity would suppress the lack of response to pheromone in the strains harboring Rheb-activating mutations. Surprisingly, this was not the case: rapamycin-treated mutant cells were still unable to respond to pheromone (control cells were slightly affected by rapamycin treatment) (Fig 6B and 6C). This result suggested that the observed negative control on mating in *U. maydis* by the presence of Rheb-activating mutations was TORC1-independent. It is worth mentioning that there is growing evidence that in eukaryotic cells (including fungi), Rheb acts using both

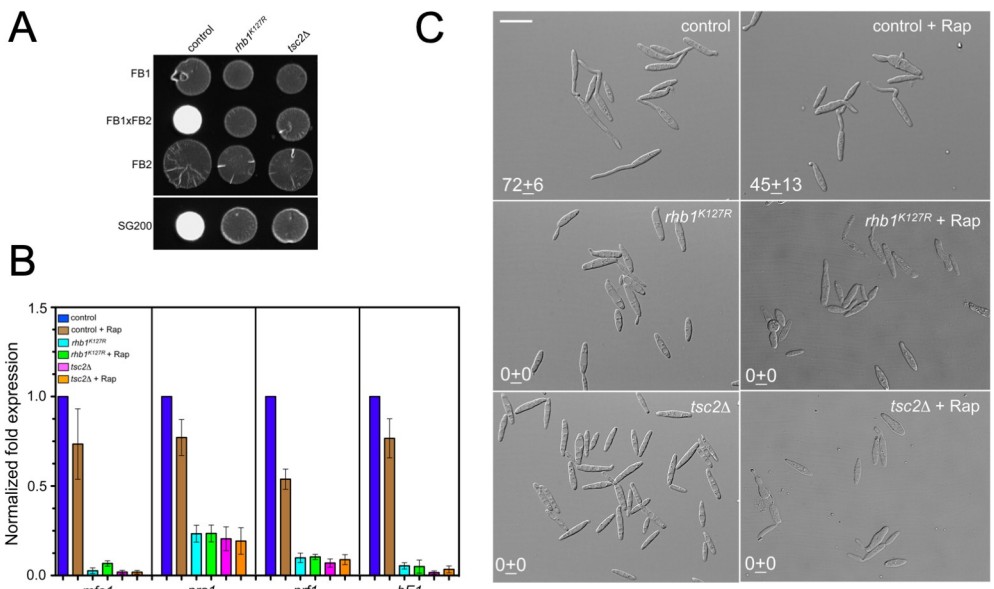

**Fig 6. Forced activation of Rheb resulted in the absence of pheromone response.** (A) Strains carrying the Rheb-activating mutations cannot form infective filaments. Control and the indicated mutant strains were spotted on minimal ammonium media charcoal plates and incubated for 24 h at 28°C. The white fuzzy colonies reflect the formation of *b*-dependent filaments. (B) qRT-PCR for the indicated genes from control (FB1) and mutant cells incubated for 6 hours in minimal ammonium media in the presence of synthetic a2 pheromone (2.5 μg/ml). Rap indicated the addition of 1μg/ml rapamycin. Values are referred to the expression of each gene in the control strain. Each column represents the mean value of three independent biological replicates. Error bars represent the SD. (C) Cell images from cultures of (B) showing the absence of conjugation tubes in mutant strains. Bar: 15 μm.

TORC1-dependent and TORC1-independent pathways [23]. We have confirmed the absence of the role of TORC1, respecting the lack of response to pheromone in the strains harboring Rheb-activating mutations, using torin1 and the *tor1*^*nar1* conditional allele (S15A and S15B Fig). Moreover, we observed that Tor1 kinase activity was somehow required for the growth of the conjugation tubes (S15C Fig)

## Rheb-activating mutations interfere with the pheromone-responsive MAPK cascade

The transcriptional activation of *a*- and *b*-mating type loci is dependent on Prf1, which directly binds the regulatory regions of these genes. Because of that, Prf1 is the main trigger of pathogenic development in *U. maydis* [14]. The transcriptional levels of *prf1* are strongly influenced by environmental conditions, including pheromone, in response to which *prf1* mRNA levels increased around 10-fold [15]. In cells harboring the Rheb-activating mutations, the transcriptional levels of *prf1* do not increase upon pheromone treatment, opening the possibility that the observed defects were linked to the inability of the mutant cells to upregulate *prf1* transcription. To address this possibility, we took advantage of a previously described *prf1* allele (*prf1*^*con*), where the *prf1* gene is highly transcribed from the constitutive *tef1* promoter, and thereby, its transcription was disentangled from environmental conditions, including the presence of pheromone [15]. We have introduced the *prf1*^*con* allele in strains carrying Rheb-activating mutations. In these strains, the expression level of the *prf1*^*con* allele was increased more than 50-fold, respecting the wild-type allele, regardless of the presence or not of pheromone (S16 Fig). However, despite the high expression of *prf1*, these strains were still unresponsive to pheromone and did not form conjugation tubes (Fig 7A).

Although environmental conditions can influence the expression levels of *prf1*, the main factor determining Prf1 activity is the protein phosphorylation mediated by the cAMP/PKA cascade and the pheromone response MAPK cascade [65,66]. While PKA sites in Prf1 are essential for induced expression of *a*- and *b*-mating-type genes, MAPK sites are not required for pheromone-induced expression of *a*-mating-type genes but are crucial for pheromone-responsive *b*-mating-type gene expression. Of note, when analyzing the transcriptional level of *a*- and *b*-mating types in the *prf1*$^{con}$ strains, we have observed that while *mfa1* and *pra1* mRNA levels were increased, the levels of *bE1* did not (Fig 7B), suggesting that Prf1 ability to activate these targets upon pheromone recognition was differentially affected by the presence of Rheb-activating mutations. One appealing explanation for such differential behavior is to assume that the cause of the absence of pheromone response in these mutants might be related to some defect in the transmission of the pheromone signal by the MAPK cascade. To address this possibility, we have introduced in the strains carrying the Rheb-activating mutations an ectopic copy of an activated allele of the MAPKK Fuz7 (*fuz7*$^{DD}$) under the control of the *crg1* promoter. The expression of this allele (upon addition of arabinose to the growth medium) bypasses the requirement of pheromone and the cascade components upstream of Fuz7 [67]. Strikingly, we observed the formation of structures resembling conjugation tubes in the Rheb-activating mutant strains expressing the *fuz7*$^{DD}$ allele. However, they were shorter, thicker, and frequently swelling at the tip (Fig 7C). Also, the expression of the *fuz7*$^{DD}$ allele resulted in mRNA levels in Rheb-activating mutant strains similar to control strains respecting *a*- and *b*-mating type and *prf1* genes (Fig 7D).

The end outcome of the pheromone cascade is the expression of the genes encoding the heterodimeric b-factor (located at the *b* locus), which activates the filamentation and the pathogenic development in *U. maydis*. We wondered whether the observed avirulence in Rheb-activating mutant strains could be ascribed to insufficient *b* expression (due to impaired pheromone response). Therefore, we introduced the Rheb-activating mutations in the solopathogenic strain HA103, which constitutively expresses an active b heterodimer [14]. We observed that the ability to produce filamentous growth in charcoal-containing plates was restored in the strains carrying the Rheb-activating, although not at similar levels as the control strain (HA103) (S17A Fig). Similarly, we observed that the plants infected with the HA103-derived cells carrying the Rheb-activating mutations developed tumors in around 45% of the cases, compared with more than 80% of plants infected with HA103 (S17B Fig).

These results strongly suggested that the virulence defect of the Rheb-activating mutants resulted -partly but not exclusively- from the attenuated expression of the *b* genes, most likely because of some defect in the transmission of the pheromone signal through the MAPK cascade, at some step upstream of Fuz7 kinase.

## The pheromone receptor seems to be mislocalized in cells carrying Rheb-activating mutations

We sought to address which elements upstream of Fuz7 kinase were affected by the presence of Rheb-activating mutations during pheromone response. Since we were adding synthetic pheromone (*i. e.*, the requirement of endogenous synthesis of pheromone was bypassed), the most upstream element of the MAPK cascade and the first candidate to be studied was the pheromone receptor Pra1 [68]. A C-terminal GFP fusion of Pra1 is fully functional, and in conjugation tubes, Pra1-GFP localizes in a cap-like manner at the growing hyphal apex [69]. We introduced the endogenous *pra1*-GFP fusion into control and Rheb-activating mutant strains and analyzed the localization of Pra1-GFP upon pheromone treatment. Pra1-GFP can be detected in control cells at the apex of the conjugation tube and vacuoles, as described [69]. In contrast, in cells carrying the Rheb-activating mutations, the GFP signal was fainter (most

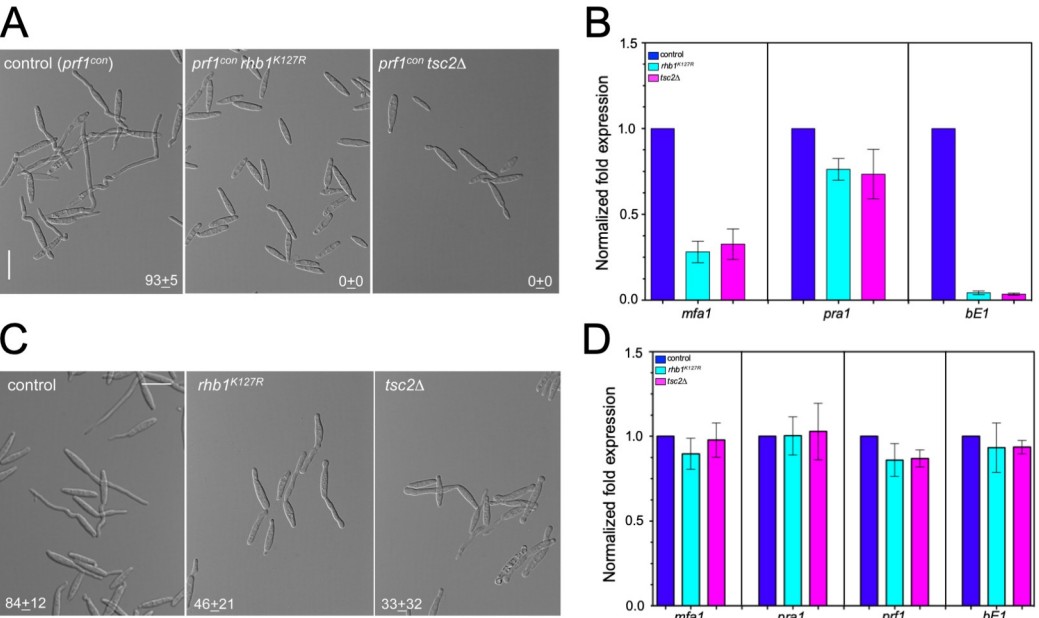

**Fig 7. Genetic activation of MAPK cascade suppressed the Rheb-dependent inhibition of the pheromone response.** (A) Cell images from cultures of strains carrying the $prf1^{con}$ allele and the indicated mutations, showing the absence or presence of conjugation tubes upon incubation for 6 hours in minimal ammonium media in the presence of synthetic a2 pheromone (2.5 μg/ml). Bar: 15 μm. The numbers indicated the percentage of cells showing conjugation tubes (mean ±SD from 3 independent experiments counting 100 cells each). (B) qRT-PCR for the indicated genes from control (FB1 $prf1^{con}$) and mutant cells incubated for 6 hours in minimal ammonium media in the presence of synthetic a2 pheromone (2.5 μg/ml). Values are referred to the expression of each gene in the control strain. Each column represents the mean value of three independent biological replicates. Error bars represent the SD. (C) Cell images from cultures of strains carrying the ectopic $P_{crg1}:fuz7^{DD}$ allele and the indicated mutations (control: FB1 $P_{crg1}:fuz7^{DD}$), showing the presence of conjugation tubes upon incubation during 6 hours in minimal ammonium medium amended with arabinose. Bar: 15 μm. The numbers indicated the percentage of cells showing conjugation tubes (mean ±SD from 3 independent experiments counting 100 cells each). (D) qRT-PCR of cultures from A. Values are referred to the expression of each gene in the control strain (FB1 $P_{crg1}:fuz7^{DD}$). Each column represents the mean value of three independent biological replicates. Error bars represent the SD.

likely because of the lack of *pra1* transcriptional induction upon pheromone treatment in mutant cells), and it was primarily located at vacuoles (Fig 8A). Because cells carrying the Rheb-activating mutations do not form conjugation tubes in response to pheromone, it was not possible to discriminate whether the lack of polar accumulation of Pra1-GFP was the cause of the absence of pheromone response (and hence of conjugation tubes), or whether the absence of conjugation tubes has as a consequence the lack of localization of Pra1-GFP signal at the apex. To address this question, we took advantage of the previous result indicating that expression of *the fuz7^{DD}* allele induces the formation of structures resembling conjugation tubes in cells harboring the Rheb-activating mutations. Strikingly, Pra1-GFP was not accumulated at the apex of these structures in cells carrying both *fuz7^{DD}* and the Rheb-activating mutations, while in control cells, it was evident an apparent accumulation of GFP signal at the tip of the conjugation tube (Fig 8B).

The decrease of the Pra1-GFP signal at the apex of the conjugation tubes in strains carrying the Rheb-activating mutations could be due to the degradation of the Pra1 receptor at the vacuole (where it seems to be accumulated). To address this possibility, we analyzed using Western blot the presence of the Pra1 receptor in control and mutant strains upon pheromone treatment (Fig 8C). We have found that even when a high percentage of the Pra1-GFP receptor seems to be degraded at the vacuole (suggested by the strong signal of the free GFP band),

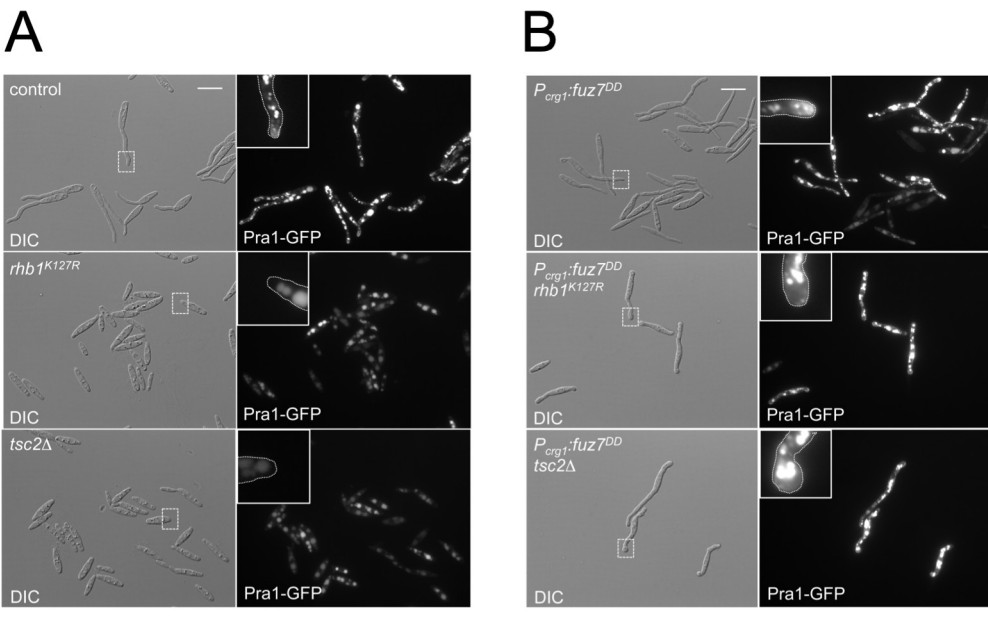

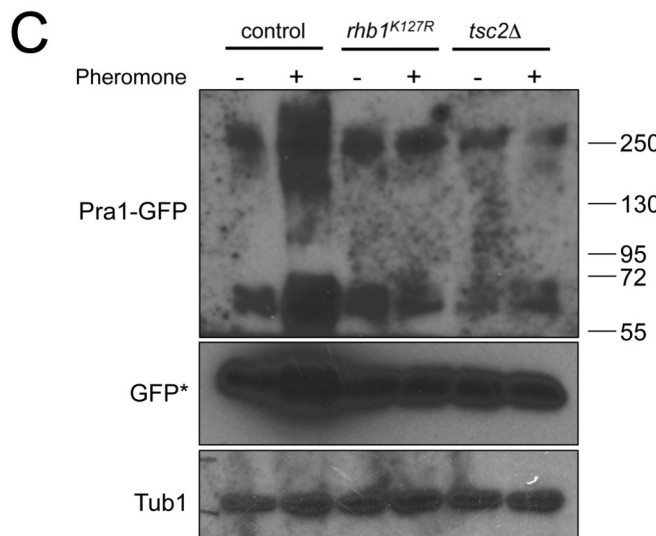

**Fig 8. Forced activation of Rheb resulted in mislocalization of the pheromone receptor.** (A) Pra1-GFP is not polarized in the strains carrying the Rheb-activating mutations. Fluorescent cell images from cultures of the indicated strains carrying an endogenous *pra1-GFP* allele incubated for 6 hours in minimal ammonium media in the presence of synthetic a2 pheromone (2.5 μg/ml). Cell images were obtained with the same time exposure. Insets showed magnification of selected cell tips (indicated in DIC images as dashed rectangles). Bar: 15 μm. (B) Pra1-GFP is not polarized in the Rheb-activating mutant strains, even in the presence of conjugation tubes. Fluorescent cell images from cultures of the indicated strains carrying an endogenous *pra1-GFP* allele and an ectopic copy of $P_{crg1}$:*fuz7$^{DD}$* transgene incubated for 6 hours in minimal ammonium media amended with arabinose to induce the expression of the activated MAPKK. Insets showed magnification of selected cell tips (indicated in DIC images as dashed rectangles). Cell images were obtained with the same time exposure. Bar: 15 μm. (C) Pra1-GFP fusion is present in conditions of forced Rheb activation. Western blot analysis of cell extracts from indicated cultures incubated for 6 hours in minimal ammonium media in the presence or absence of synthetic a2 pheromone (2.5 μg/ml). The same filter was incubated in succession with anti-GFP and anti-tubulin (as loading control). The filter incubated with anti-GFP was exposed at different times (GFP* exposition was 1/10 from Pra1-GFP exposition). The expected size of Pra1-GFP fusion was 67,63 kD, but a significant fraction migrates at much higher sizes (ca. 250 kD) in all cases.

there is also a significant signal associated with full-length Pra1-GFP receptor in all conditions and strains. From these results, we concluded that the degradation of the Pra1 receptor is unlikely to be the cause of the absence of polar localization of Pra1-GFP in strains carrying the Rheb-activating mutations. Hence, we favored the hypothesis that the misleading signaling induced by the forced Rheb activation somehow interferes with the correct localization of Pra1. Because previous studies indicated that the lack of accumulation of Pra1 receptor to the apex of the cell resulted in an impaired pheromone signal transmission [69], we wish to propose that the apparent mislocalization of Pra1 is the cause of pheromone insensitivity of strains carrying the Rheb-activating mutations.

## The ubiquitin ligase Rsp5 and the α-arrestin Art3 are required for the pheromone insensitivity that results from Rheb activation

We sought to add further support to our hypothesis about the mislocalization of Pra1 as the origin of the lack of virulence when Rheb is inappropriately activated. In *S.pombe*, Rheb negatively regulates distinct proteins, which destiny is the PM (such as amino acid transporters), either by promoting its degradation or inhibiting their transport to PM. Some of these activities are TORC1-dependent while others are not, but the ubiquitin ligase Pub1/Rsp5 mediates all, often associated with α-arrestins [52,61,70]. We observed that the absence of the proposed Rsp5 ubiquitin ligase restitutes the ability of strains carrying the Rheb-activating mutations to produce infective filaments upon mating, assessed as a positive *fuz* phenotype on charcoal plates (Fig 9A). It is worthy of mention that *rsp5Δ* alone has some negative effect on *fuz* phenotype since the apparent density of white filaments on charcoal was lower respecting control cells (see below for additional defects of *rsp5Δ*). Encouragingly, we also observed that deletion of *art3*, encoding one of the proposed α-arrestins in *U. maydis*, totally restitutes the *fuz* phenotype to crosses of strains carrying either the *rhb1^{K127R}* or *tsc2Δ* alleles (Fig 9A). We addressed whether the inactivation of either Rsp5 or Art3 resulted in the formation of conjugation tubes in response to pheromone and whether this restitution correlated with the Pra1-GFP accumulation at the tip apex. We found this to be the case (Fig 9B). We observed that the conjugation tubes showed altered morphology, characterized by bulges or additional growth points at the filament base, although we believe that these defects were attributable to the lack of function of Rsp5 and Art3 because we observed similar defects in the single *rsp5Δ* or *art3Δ* mutants (S18 Fig). We also analyzed the transcriptional response to pheromone respecting genes belonging to *a*- and *b*-loci, and coherently, this response was also restituted (S19 Fig). Finally, we sought to address whether the conjunction of the Rheb-activating mutations and the loss-of-function of Rsp5 or Art3 restituted the infectivity of *U. maydis* mutant cells. Firstly, we analyzed the infectivity of strains carrying the *rsp5Δ* or *art3Δ* mutations alone. We found that cell crosses carrying *rsp5Δ* resulted in avirulence (S20 Fig), while crosses of *art3Δ* strains resulted in plant infections in a significant frequency (around 70% of the plants). However, the more severe disease symptom found was the formation of small tumors (Fig 9C). We carried out infection with crosses from strains harboring the double combination *art3Δ* and either the *rhb1^{K127R}* or *tsc2Δ* alleles, and we have found that in these combinations, the distinct strains were also able to infect plants and showed similar defects as *art3Δ* alone (Fig 9C and 9D).

In summary, we have found that disabling Rsp5 and Art3 released the inhibitory effect in the pheromone response by the presence of Rheb-activating mutations.

## Discussion

Our initial aim in this study was to address whether inappropriate activation of TORC1 disabled the ability of *U. maydis* to infect plants. For that, we seek to activate the TORC1 complex

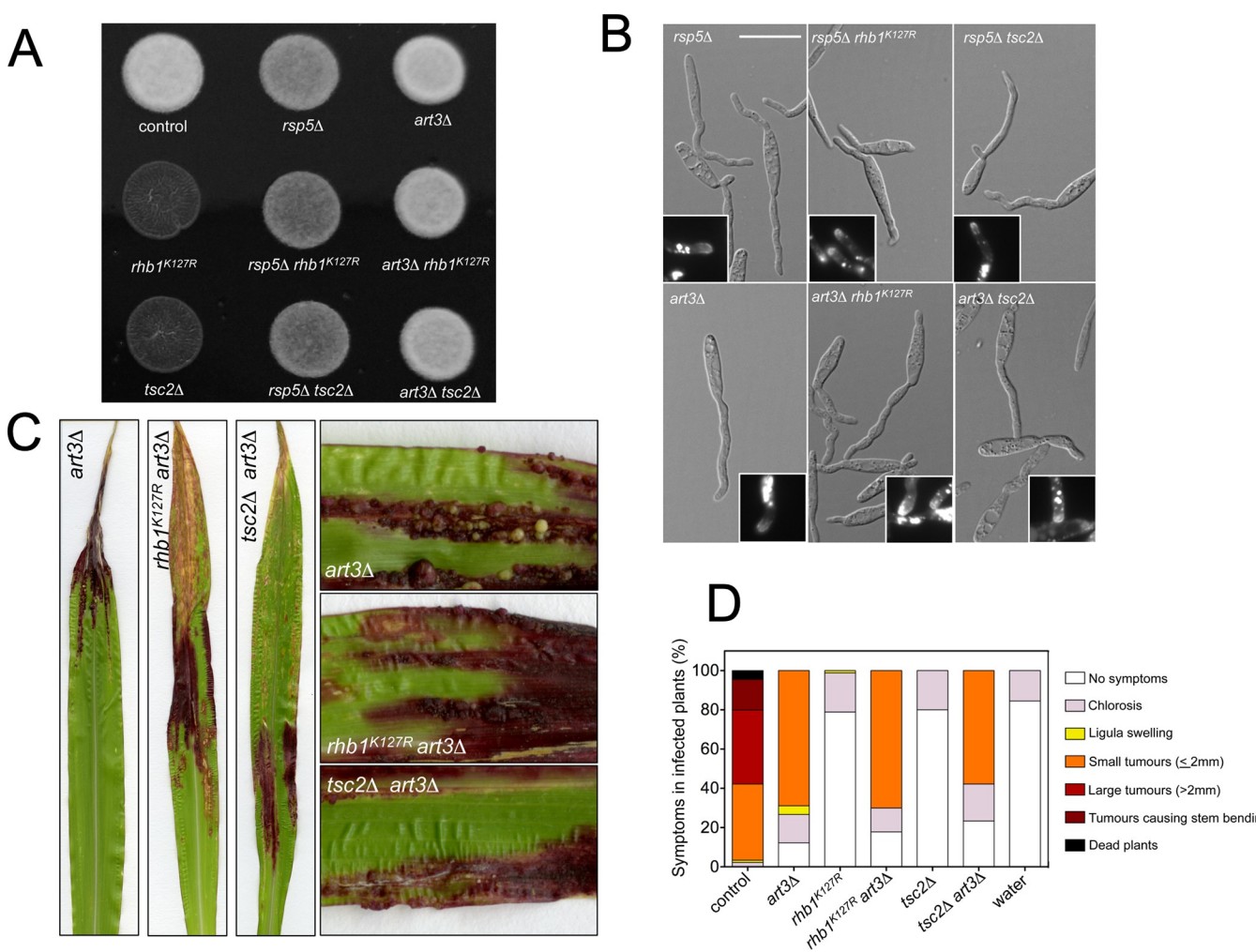

**Fig 9. The inhibition of the pheromone response by unscheduled activation of Rheb requires Rsp5 and Art3.** (A) Strains carrying the Rheb-activating alleles and loss-of-function mutations in *rsp5* or *art3* were able to form infective filaments. Crosses of control (FB1xFB2) and the indicated mutant strain combinations were spotted on minimal ammonium media charcoal plates and incubated for 24 h at 28°C. The white fuzzy colonies reflect the formation of *b*-dependent filaments. (B) Strains carrying the Rheb-activating alleles and loss-of-function mutations in *rsp5* or *art3* were able to form conjugation tubes and polarize Pra1-GFP. Fluorescent cell images from cultures of the indicated strains carrying an endogenous *pra1-GFP* allele incubated for 6 hours in minimal ammonium media in the presence of synthetic a2 pheromone (2.5 µg/ml). Cell images were obtained with the same time exposure. Insets showed magnification of selected cell tips. Bar: 15 µm. (See S19 Fig for wide-field images). (C) Strains carrying the Rheb-activating alleles and loss-of-function mutations in *art3* were able to infect plants. Representative leaves 14 days after infection with the indicated strains is shown. (D) Graph showing disease symptoms caused by crosses of wild-type (control) and the indicated mutant strains. The symptoms were scored 14 days after infection. Three independent experiments were carried out, and the average values are expressed as a percentage of the total number of infected plants (n: 30 plants in each experiment).

directly. We took advantage of work done in fission yeast, where it was defined that two classes of mutations led to the overactivation of Rheb, one of the GTPase complexes involved in the promotion of TORC1 activity: point mutations in the Rheb GTPase [47,71] and loss-of-function of its cognate GAP, Tsc2 [72,73]. We recreated the Rheb-activating mutations in our fungal system and demonstrated that these mutations' presence resulted in a TORC1 activation in *U. maydis*. Encouragingly, we also observed that cells carrying the Rheb-activating mutations were unable to infect plants. However, much to our surprise, the observed inhibition of the virulence was TORC1-independent.

The TSC-Rheb regulatory axis has been extensively characterized in metazoan, which conveys developmental signals and energy and oxygen levels [74]. Although the vast majority of

research on these proteins has focused on their interaction with TORC1, it is well established that the TSC-Rheb axis has functions independent of TORC1 [23,75]. Unfortunately, except for *S. pombe*, Rheb and TSC have been poorly characterized in fungi. In *S. pombe*, Rheb plays a central role in regulating the import of nutrients by remodeling their proteome at the PM, including nutrient transporters or permeases [76]. For that, Rheb promotes (via TORC1-dependent and TORC1-independent pathways) the interaction of the nutrient permeases with the HECT-type ubiquitin ligase Pub1(Rsp5) and α-arrestin-like proteins (ART), a family of ubiquitin ligase adaptors [62]. The outcome of these interactions is variable, ranging from the displacement out of the PM to the vacuole degradation of the specific permease [52,59,61]. We have found that in *U. maydis*, Rheb-activating mutations resulted in TORC1-dependent down-regulation of the nitrate permease (Nrt1), even in the presence of nitrate as the sole nitrogen source. This down-regulation likely resulted from the permease degradation by the vacuole, and it was dependent on Rsp5. We believe that some ART protein could also be involved in this regulation, although we cannot provide any evidence. We propose three possible reasons for our failure. We identified four putative ART proteins (Art1-4), although we could obtain loss-of-function alleles from only three (Art1-3). One possibility is that the putative ART involved in degrading Nrt1 was the one encoded by *art4*. Also, we cannot discard that in *U. maydis*, additional ART proteins were present. The arrestin domains, typical signatures of ART proteins, are not easily predicted from the primary sequence because they are interspersed at varying positions by large, probably unstructured loops, which modulate the function and location of ARTs in ways that are currently only partially understood [62]. Finally, some ART–Rsp5 complexes function partially redundant with overlapping substrate specificity [62], and we have analyzed single mutations and not combinations of them.

Using the Rheb-activating mutations, we also found that the TSC-Rheb axis negatively controlled the response to pheromone, most likely acting on the cellular distribution of the pheromone receptor. In response to pheromone, the Pra1 receptor accumulated at the cell tip, and this accumulation was required for pheromone perception and, consequently, for conjugation tube formation [69]. We observed that in Rheb-activating mutations, the Pra1-GFP fusions do not accumulate at the polarized tip, even in conditions bypassing the pheromone transmission cascade by expressing a constitutively active Fuz7 MAPK. The apparent inhibition of the Pra1-GFP polarized accumulation required Rsp5 and the Art3 α-arrestin-like protein, although it was TORC1-independent. We do not address the details of the molecular mechanism involved in this down-regulation. In *S.pombe*, activated Rheb promoted the interaction between Pub1(Rsp5) and the ART protein Any1. The complex prevented the localization of the amino acid permease Aat1 at the PM by a still unknown mechanism without the involvement of TORC1 activity [52]. We have tried to study whether Rheb-activating mutations promoted the interaction between Rsp5 and Art3 in *U. maydis*. However, the tagging of Rsp5 (either at the N- or C-terminal end) inactivated the protein. Despite this negative result, we believe that, as described in other systems [62], Rsp5 and Art3 most likely could be acting as complex. In *S. cerevisiae*, Ste3 (the closest homolog to Pra1) is down-regulated as part of the pheromone adaptation process (independent of TORC1). This action involves the concourse of Rsp5 and three redundant arrestins (Art1/Lbd19, Art3/Aly2, and Art6/Aly1) [77,78]. *U. maydis* Art3 was the closest homolog to Art3 and Art6 from *S. cerevisiae* (we did not find a homolog to *S. cerevisiae* Art1, see S13 Fig), suggesting some evolutionary conservation. The proposed mechanisms for this down-regulation in *S. cerevisiae* include the transport of the receptor via the Golgi body to vacuole more efficiently than delivered from the Golgi compartment to the PM and an increase of the receptor endocytosis. For *U. maydis*, the transport pathways by which the Pra1 receptor reaches the tip of the conjugative tube are currently

unknown. It deserves future studies to understand at which level the Rheb-activating mutations interfere with such transport.

The final purpose of pheromone signaling is the assembly of the heterodimeric b-factor, the transcriptional activator that lies at the top of the transcriptional program leading plant infection. For that, the pheromone cascade is responsible for expressing the corresponding b-encoding genes in each mating partner and promoting the recognition and fusion of the distinct sexual cells. Coherently with our hypothesis that Rheb is interfering with the pheromone signaling, we have found that the forced expression of the compatible b subunits in the HA103 solopathogenic strain (and thereby, bypassing the requirement of the pheromone cascade) suppressed the effects of Rheb-activating mutation respecting the ability to produce filamentous growth in charcoal-containing plates and virulence. However, we also observed that this suppression was partial, strongly suggesting the presence of additional targets of Rheb involved in virulence and filamentous growth, most likely downstream of the b-factor. These putative targets will deserve future attention.

We also found that although the *rsp5* mutation restituted the pheromone transmission in cells carrying the Rheb-activating mutations, it was apathogenic. Rsp5 has been involved in essential roles in processes apart from nutrient transport control in other organisms. One of these roles was related to DNA-damage responses and the degradation of the mitotic activator Cdc25 [79,80]. The mandatory G2 cell cycle arrest occurring during the formation of the infective filament in *U. maydis* involves the DNA-damage cascade and the degradation of Cdc25 by a still unknown mechanism [81,82]. We have observed that cells carrying *rsp5* mutation alone were affected in the formation of infective filament, opening the possibility that Rsp5 could be involved in this process. In the same way, we also observed that a*rt3*Δ mutants showed decreased symptom development after plant infection. Interestingly, these defects were reminiscent of the consequences of defective hexose transporters in *U. maydis* during biotrophic growth [83]. ART proteins have also been involved in the functionality of sugar transporters [62], and arrestins were described to have roles during the biotrophic development of *M. oryzae* [84]. All these observations about the defects associated with *art3* and *rsp5* single mutants merit further studies.

Besides these opening questions, we believe that the most striking observation in our study was that TORC1 seemed not to be involved in the negative regulation of virulence in *U. maydis*. This observation contrasts with previous studies in other phytopathogenic fungi, where it was proposed that the surveillance of environmental conditions (mainly nutrients and surface cues) modulates the virulence program by controlling TORC1 activity [5,6,9,11]. The TORC1 targets involved in virulence are currently unknown. Studies in *M. oryzae* have demonstrated that low TORC1 activity during spore germination on inductive nutrient-free, hydrophobic surfaces is required to activate a sort of metabolic checkpoint that temporarily halts the cell cycle at the G2 phase [8]. This cell cycle arrest induces both appressorium formation and autophagy [85]. In *U. maydis*, a G2 cell cycle arrest is associated with the formation of the conjugation tube and the infective filament. However, the mechanisms for this arrest do not involve the activity of TORC1. They are mediated by the pheromone cascade-promoted phosphorylation of an importin [81,82] and the b factor-dependent activation of the DNA-damage cascade [86,87]. In other words, in *U. maydis*, TORC1 is not required to be down-regulated for the cell cycle arrest occurring during the formation of virulence structures. On the contrary, we suspect that TORC1 activity could be required to form the infective structures in *U. maydis*. The development of conjugation tubes implies a fast and robust polar growth, which most likely requires the concourse of the TORC1 pathway, a well-established promotor of growth in eukaryotes. Supporting our suspicion, we have observed that in cells in which TORC1 activity was inhibited during the pheromone response, the length of conjugation tubes seemed to be

affected (S15C Fig). However, the transcriptional program associated with pheromone response was activated (S15B Fig). We believe that because of this proposed requirement, TORC1 activity has been disentangled from virulence control in *U. maydis*. Taking into account all these considerations, together with the possibility of the TSC-Rheb axis to signal by TORC1-dependent and -independent ways, we wish to propose that in *U. maydis*, environmental situations precluding the activation of the virulence program would impinge on the activation of the TSC-Rheb axis, which could result in blindness to pheromone via the mislocalization of the pheromone receptor. We think this could be an efficient way to save cell resources, halting the entire pathogenic development at the very early stages (S21 Fig).

We wondered whether the involvement of Rheb and its regulation by the TSC complex in the control of virulence is a unique feature of *U. maydis* or can be translated to other phytopathogenic fungi. Rheb and its regulation by the TSC complex have been poorly characterized in fungi. In *Aspergillus fumigatus* and *Candida albicans*, loss-of-function mutants in Rheb affect the response to nitrogen-starvation processes and the ability to grow, hence the ability to infect animals [88–90]. To the best of our knowledge, the only report connecting Rheb regulation and virulence concerns to the mycoparasite *Trichoderma atroviride*, where loss-of-function mutants in the Rheb GTPase were as effective as the wild type in attacking the plant pathogen *Rhizoctonia solani*, while loss-of-function in Tsc2 resulted in impaired mycoparasitism (although authors do not show whether this effect was TORC1- dependent or not) [91]. We wish to propose that the TSC-Rheb axis could be an alternative manner to transmit environmental conditions in phytopathogenic fungi by TORC1-dependent and -independent pathways. A deeper characterization of this pathway in phytopathogenic fungi, and more concretely its unappropriated activation, would support its use as a target of actuation to design antivirulence therapies. The TSC-Rheb axis is absent in plants [16], and the activation of Tor in plants requires a GTPase from the Rho family called Rop2 [92], a different family from Rheb-like GTPases. This could facilitate the search for chemicals acting specifically in Rheb-like GTPases without interference in plant TOR metabolism.

## Material and methods

### *U. maydis* growth conditions

*Ustilago maydis* strains are derived from FB1, FB2, and SG200 genetic backgrounds [63,64] and are listed in S2 Table. Cells were grown in rich medium (YPD) or minimal medium (MMD) [93]. Controlled expression of genes under the *crg1* and *nar1* promoters was performed as described previously [24]. Rapamycin (0.1 mg/ml) and Torin1 (1mM) stocks were dissolved in DMSO as solvent. In control plates, DMSO was added to 1%.

### Plasmid construction and *U. maydis* strain generation

*U. maydis* DNA isolation was performed as previously described [94]. To construct the different strains, the transformation of *U. maydis* protoplasts with the desired constructions was performed [94]. Integration of the corresponding construction into the corresponding loci was verified in each case by diagnostic PCR and subsequent Southern blot analysis or RT-PCR analysis of transcripts depending on the type of integrated mutant allele.

Plasmid pJET1.2 (Thermo Fisher) was used to subclone and sequence genomic fragments generated by PCR.

Plasmids for conditional alleles of *tor1*, *rpt1*, and *rhb1*; C-terminal tagging of endogenous loci from *aga1*, *maf1*, *nrt1*, and *pra1* with 3HA or GFP; and loss-of-function alleles of *sin1*, *rct1*, *sch9*, *aga1*, *tsc2*, *nrt1*, *fpr1*, *rsp5*, *art1*, *art2*, *art3*, and *art4* were performed using Golden Gate assembly. Genomic PCR fragments flanking the interest cassettes were amplified and

assembled following published procedures [95]. Mutant allele *prf1$^{con}$* was already described [15]. Atg8 tagged at the N-terminus with GFP was constructed by inserting the corresponding mutant cassette associated with a resistance to hygromycin flanked by FRT sites to be removed upon transformation with a plasmid encoding a FLPase recombinase [96]. Further details of the constructions above and sequences of the plasmids are available on request.

For ectopic expression, plasmids carrying the respective alleles were inserted at the *ip* locus [24]. The mutant allele *P$_{crg1}$:fuz7$^{DD}$* was already described [67]. To construct the ectopic expression of *rhb1$^{K127R}$*, a synthetic DNA fragment of 676 bp, carrying the desired mutation (associated with an *Afe*I restriction site for diagnostic purposes) and flanked by *Nde*I and *Eco*RI sites were cloned into the respective sites of plasmid pRU2 [24].

The *rhb1$^{K127R}$* endogenous mutant allele was constructed using the published CRISPR-Cas9 *U. maydis* system [97]. The plasmid pCas9Rhb1KR was generated by insertion using Gibson assembly of a 177 bp DNA fragment carrying the template for the sgRNA synthesis (see the nucleotide sequence in S1 Methods) into the unique *Acc*65I site from pCas9_sgRNA_0. As a DNA repair template, a synthetic DNA fragment of 676 bp carrying the desired mutation (associated with an *Afe*I restriction site for diagnostic purposes) as well as a single mutation eliminating the PAM site (see the nucleotide sequence in S2 Methods) was co-transformed with the pCas9Rhb1KR plasmid as described [98].

## RNA analysis

Total RNA was extracted with an acidic phenol solution. After extraction, the RNA was cleaned using the High Pure RNA Isolation Kit (Roche Diagnostics GmbH). cDNA was synthesized using the High Capacity cDNA Reverse Transcription Kit (Applied Biosystems), employing 1 μg total RNA per sample. qRT-PCR was performed using the SsoAdvanced Universal SYBR Green Supermix (BioRad) in a CFX96 Real-Time PCR system (BioRad). Reaction conditions were as follows: 3 min 95˚C followed by 40 cycles of 10 sec 95˚C/10 sec 60˚C/30 sec 72˚C. The expression of *tub1* (encoding Tubulin α) was used as an internal control.

A list of the primers used for qRT-PCR can be found in S3 Table.

## Cell lysates preparation and gel electrophoresis analyses

Denatured protein extracts were performed using an adapted chloroacetic acid (TCA) method. Briefly, cells from 5- to 10-ml aliquots of cultures were harvested, and 1 ml of 20% TCA was added. The supernatant was removed after centrifugation, and the pellet was resuspended in 100 μl of 20% TCA and stored at −80˚C. Samples were thawed on ice, glass beads were added, and cells were broken using a FastPrep FP120 cell disrupter (BIO 101 ThermoSavant, Obiogene, Carlsbad, CA). The lysate was recovered by punching a hole in the bottom of the tube, and the glass beads were further washed with 200 μl of 5% TCA. Lysates were centrifuged at 1000×*g* for 3 min, and the pellet was thoroughly resuspended in 100 μl of 2×Laemmli buffer and 50 μl of 2 M Tris base. After boiling for 5 min, 10–20 μl from each sample was loaded in the gels. TGX (4–20%) gels from BioRad were used at constant 100v running conditions.

To detect the phosphorylated forms of Maf1-3HA and Aga1-3HA, crude protein extracts were prepared and submitted to immunoprecipitation. Briefly, cells were harvested by centrifugation at 4˚C and washed twice with ice-cold water. The cell pellet was resuspended in ice-cold HB buffer (25 mM MOPS pH 7.2, 15 mM MgCl2, 15 mM EGTA, 1% Triton X-100, PhosSTOP [1 tablet per 10 mL], Roche protease inhibitor cocktail [1 tablet per 10 mL]) and cells were broken using a FastPrep FP120 cell disrupter. The lysate was recovered by punching a hole in the bottom of the tube, the glass beads were further washed with ice-cold HB buffer, and the lysate was cleared by centrifugation (10 min/13000xg). For immunoprecipitation

analysis, approximately 3.5 mg of total protein extracts (1 ml) were incubated with 1 μg of the anti-HA monoclonal antibody for 2 hours at 4°C and then prewashed G-protein coupled magnetic beads (50 μl) were added and incubated for 30 minutes at 4°C with agitation. Immunoprecipitates were washed six times with 1 ml of HB buffer. To resolve the phosphorylated forms of Maf1-3HA and Aga1-3HA, 8% gels (acrylamide:bisacrylamide 29:1) containing 30.5 μM and 45 μM Phos-tag (Wako Chemicals) respectively and 75 μM MnCl2 were used. Gels were run on ice at 100 volts in a MiniProtean3 (Bio-Rad) for 3 h. After running, gels were washed with 1 mM EDTA before transfer to PVDF membranes. To detect the tagged proteins, commercial antibodies were used. The primary antibody was followed by a secondary antibody conjugated to horseradish peroxidase, and immunoreactive proteins were visualized using a chemiluminescent substrate. The chemiluminescent signal was captured with a Fusion FX6 system (Vilber) and quantified with the Evolution-Capt software (Vilber). Each Western blot analysis was performed from at least three independent experiments.

### Plant infections, mating assays

Pathogenic development of wild-type and mutant strains was assayed by plant infections of the maize (*Zea mays*) variety Early Golden Bantam (Olds seeds) as described before [99]. For mating assays, strains were crossed on charcoal-containing minimal ammonium medium plates and incubated at 22°C [93].

### Microscopy

Images were obtained using a Nikon Eclipse 90i fluorescence microscope with a Hamamatsu Orca-ER camera driven by Metamorph (Universal Imaging, Downingtown, PA). Images were further processed with ImageJ software.

### Quantification and statistical analysis

To determine the statistical significance of differences, a two-tailed Student *t*-test was used. *P*-Values were calculated with the GraphPad Prism 5.0 software.

### Supporting information

**S1 Fig. TORC1 conditional alleles.** (A and C) schemes of *tor1^nar1* and *rpt1^nar1* conditional alleles. (B and D) qRT-PCR for *tor1* and *rpt1* mRNA levels from control and *tor1^nar1* or *rpt1^nar1* conditional cells, incubated for 8 hours in YPD (repressive conditions) or in nitrate minimal medium (MMD, permissive conditions). Values are referred to the expression of *tor1* or *rpt1* in FB1 (control strain) grown in YPD. Each column represents the mean value of three independent biological replicates. Error bars represent the SD.
(TIF)

**S2 Fig. Inhibition of TORC1 resulted in vacuole enlargement.** Representative images of liquid cultures of cells carrying the indicated conditional allele or treated with 10 μM Torin1 and 1μg/ml rapamycin (control: FB1), incubated in YPD for 8 hours. Fluorescent images showed FM4-64 staining. Bar: 15 μm.
(TIF)

**S3 Fig. TORC1 inhibitors affect cell growth.** Growth curves of liquid YPD cultures of FB1 cells amended with the indicated concentrations of Torin1 or rapamycin. Control conditions included 1% DMSO as solvent.
(TIF)

**S4 Fig. TORC2 mutants.** (A) Representative images of liquid cultures of cells carrying the indicated mutant allele incubated in YPD for 6 hours. Bar: 17 μm. (B) serial tenfold dilutions of cultures from strains carrying the indicated mutant alleles (control: FB1), spotted in solid YPD. Plates were incubated for 3 days at 28˚. C, cell length of TORC2 mutant cells. The graph shows the result from three independent experiments, counting more than 50 cells. $^{**}$p$<$0.01 based on a two-tailed Student´s $t$-test to control sample.
(TIF)

**S5 Fig. C-terminal ends of Rps6 in fungi showing the conserved phosphorylation site for S6K.**
(TIF)

**S6 Fig. Cladogram of AGC kinases described as TOR pathway targets from fission (Sp) and budding (Sc) yeast including the two predicted AGC kinases from *U. maydis*.** The tree was created by a neighbor-joining analysis without distance corrections using ClustalW. The scale bar denotes substitutions per site. The AGC association to TORC1 or TORC2 is indicated in the cases of budding and fission yeast. Note that in *U. maydis*, Aga1 was included in the group of TORC2 effectors, while Sch9 was included in the TORC1 effectors.
(TIF)

**S7 Fig. Analysis of phosphorylated Aga1-3HA.** (A) Our first attempts to discriminate phosphorylated forms of Aga1-3HA were made using BioRad TGX (4–20% acrylamide) gels from TCA samples (left panel). Although a faint band shift was observed, it was not consistently reproduced. Using Phos-tag gels (right panel) improved the differential electrophoretic mobility. We have tried both TCA extracts and immunoprecipitated samples, and we have found that immunoprecipitated samples resulted in the more reproducible and consistent data. (B) Western blot of 5 microliters from crude extract (WCE) or immunoprecipitated (IP) samples extracted from cultures carrying the Aga1-HA allele grown on YPD or minimal medium amended with nitrate (MMD). The samples were loaded in BioRad TGX (4–20% acrylamide) gels.
(TIF)

**S8 Fig. Analysis of phosphorylated Maf1-3HA.** (A) Protein alignment showing the region of similarity between the predicted *U. maydis* Maf1 and the pfam indicative of Maf1 superfamily. (B) electrophoretic mobilities of Maf1-3HA samples separated in TGX (TCA samples) or Phos-tag gels (immunoprecipitated samples). (C) Western blot of 5 microliters from crude extract (WCE) or immunoprecipitated (IP) samples extracted from cultures carrying the Maf1-HA allele grown on YPD or minimal medium amended with nitrate (MMD). The samples were loaded in BioRad TGX (4–20% acrylamide) gels.
(TIF)

**S9 Fig. Ectopic expression of *rhb1*$^{K127R}$ allele under *nar1* promoter.** (A) Scheme for the integration of the plasmid harboring the ectopic *rhb1*$^{K127R}$ allele under *nar1* promoter. Integration takes place at the *ip* locus (conferring resistance to carboxin). (B) qRT-PCR of *rhb1* mRNA levels from control and strains carrying the indicated ectopic alleles, incubated for 8 hours in YPD (repressive conditions) or in nitrate minimal medium (MMD, permissive conditions). Values are referred to the expression of *rhb1* in FB1 (control strain) grown in YPD. Each column represents the mean value of three independent biological replicates. Error bars represent the SD.
(TIF)

**S10 Fig. Growth defects of Rheb-activating mutants.** (A) Cells carrying mutations that upregulated Rheb were affected in growth in minimal medium amended with nitrate. Serial tenfold dilutions of cultures from the indicated strains (control: FB1), spotted in solid yeast-peptone medium (YPD, restrictive conditions for P$_{nar1}$ expression) or solid nitrate minimal medium (MMD, permissive conditions for P$_{nar1}$ expression), with or without 1μg/ml rapamycin. Plates were incubated for 3 days at 28˚C. (B) Ectopic expression of *rhb1*$^{K127R}$ allele under *crg1* promoter. qRT-PCR of *rhb1* mRNA levels from control and strains carrying the indicated ectopic alleles, incubated for 8 hours in YP (yeast-peptone) or in nitrate minimal medium amended with glucose (repressive condition) or arabinose (permissive condition). Values are referred to the expression of *rhb1* in FB1 (wt strain) grown in YPD. Each column represents the mean value of three independent biological replicates. Error bars represent the SD. (C) The quality of the nitrogen source influences the growth of strains expressing an overactivated Rheb. Serial tenfold dilutions of cultures from the indicated strains (control: FB1 P$_{crg1}$:*rhb1*), spotted in solid yeast peptone medium or minimal nitrate medium amended either with arabinose (YPA or MMA, permissive conditions for P$_{crg1}$ expression), or glucose (YPD or MMD, restrictive conditions for P$_{crg1}$ expression) as carbon source. Plates were incubated for 3 days at 28˚C. (TIF)

**S11 Fig. Characterization of *nrt1*, encoding the *U. maydis* nitrate permease.** (A) Cells lacking *nrt1* were unable to grow in a minimal medium using nitrate as the sole nitrogen source. Serial tenfold dilution of cultures from the indicated strains (control: FB1), spotted in solid YPD, ammonium minimal medium and nitrate minimal medium. Plates were incubated for 3 (YPD) and 4 (minimal medium) days at 28˚. (B) the expression of *nrt1* was explicitly induced by the presence of nitrate in the medium. qRT-PCR of *nrt1* mRNA levels from FB1 cells incubated for 8 hours in the indicated liquid medium. Values are referred to the expression of *nrt1* with respect the expression of *tub1* (encoding Tubulin α) in the respective growth medium. Each column represents the mean value of three independent biological replicates. Error bars represent the SD. (C) Nrt1-GFP fusion was located at PM depending on the presence of nitrate as the nitrogen source. Fluorescence images of cultures from a strain carrying a *nrt1-GFP* endogenous allele. Cells were grown for 8 hours in YPD or minimal nitrate medium. A magnification to show the plasmatic membrane accumulation of the fluorescence can be seen in the inset. Images from YPD medium were exposed 6 times more with respect images from minimal medium. Bar: 10 μm. (D) Western blot analysis of cell extracts with anti-GFP antibodies from a strain carrying a *nrt1-GFP* endogenous allele grown in different media for 8 hours. Levels of Tub1 were used as loading control (bottom blot). In addition to a band corresponding to the expected size of the Nrt1-GFP fusion, we had found a strong signal corresponding to the size of GFP alone. This signal can be explained by assuming that the Nrt1-GFP fusion protein was hydrolyzed in the vacuole (the GFP half is not sensitive to the proteases resident in vacuoles), which is compatible with the proposed mechanism of permeases recycling using the endocytic/vacuole pathway observed in other fungi. Addition of rapamycin (Rap, 1μg/ml) does not alter this pattern.
(TIF)

**S12 Fig. The FKBP12 homolog from *U. maydis* is required for Rapamycin-mediated suppression.** (A) Serial tenfold dilutions of cultures from FB1 (control) and a strain lacking *fpr1* (UMAG_11054, encoding the FKBP12 protein) spotted in YPD amended with 1μg/ml rapamycin. Plates were incubated at 28˚C for 2 days (control) and 3 days (rapamycin). Control plates included DMSO (1% final) as solvent. (B) Fluorescence images of cultures from the indicated strains (control: FB1 P$_{nar1}$: *rhb1*$^{K127R}$ *nrt1-GFP*; fpr1Δ: FB1 P$_{nar1}$: *rhb1*$^{K127R}$ *nrt1-GFP* *fpr1Δ*) grown for 8 hours in a minimal nitrate medium amended with 1μg/ml rapamycin. Bar:

20 μm. (C) Line-scan analysis of Nrt1-GFP signal intensities from B. 5 cells from each strain were scanned through its medial zone. Arrows indicated the location corresponding to the plasma membrane. A. U., arbitrary units.
(TIF)

**S13 Fig. Arrestins from _U. maydis_.** (A) The cladogram was created by a neighbor-joining analysis without distance corrections using ClustalW. The scale bar denotes substitutions per site. (B) Serial tenfold dilutions of cultures from the indicated strains (control: FB1 P$_{nar1}$:$rhb1$), spotted in solid yeast-peptone medium (YPD, restrictive conditions for P$_{nar1}$ expression) or solid nitrate minimal medium (MMD, permissive conditions for P$_{nar1}$ expression). Plates were incubated for 3 days at 28˚C.
(TIF)

**S14 Fig. _U. maydis_ mutants carrying Rheb-activating alleles.** (A) Cells from FB1- FB2- and SG200-derived strains carrying endogenous alleles that up-regulated Rheb1 were spotted in serial tenfold dilutions in YPD, ammonium minimal medium and nitrate minimal medium. The plates were incubated at 28˚C for 3 days. (B) Analysis of TORC1 activity using the electrophoretic mobility of Maf1-3HA as readout. Indicated strains were grown for 8 hours in ammonium minimal medium and anti-HA immunoprecipitates from cell extracts were separated in Phos-tag gels and subjected to immunoblot analysis with anti-HA.
(TIF)

**S15 Fig. Down-regulation of Tor1 activity does not suppressed the Rheb-dependent inhibition of the pheromone response.** (A) Cell images from cultures of strains carrying the indicated mutations (control: FB1) showing the absence or presence of conjugation tubes upon incubation during 6 hours in minimal ammonium media in presence of synthetic a2 pheromone (2.5 μg/ml). Torin1 was added to 10μM. Bar: 20 μm. (B) qRT-PCR for the indicated genes from control (FB1) and mutant cells incubated for 6 hours in minimal ammonium media in presence of synthetic a2 pheromone (2.5 μg/ml). Torin1 was added to 10μM. Values are referred to the expression of each gene in control strain. Each column represents the mean value of three independent biological replicates. Error bars represent the SD. (C) Tor1 kinase seems to be required for appropriated growth of the conjugation filaments. Control (FB1), torin1 (FB1 incubated 10μM torin1) and tor1$^{nar1}$(FB1 $tor1^{nar1}$) cells were incubated during 6 hours in minimal ammonium media in presence of synthetic a2 pheromone (2.5 μg/ml). The length from 50 cells (from 2 independent experiments) was measured and plotted.
(TIF)

**S16 Fig. Expression levels of _prf1$^{con}$_.** qRT-PCR of _prf1_ mRNA levels from control (wt, FB1 cells) and strains carrying the indicated alleles, incubated for 6 hours in minimal ammonium media in presence of synthetic a2 pheromone (2.5 μg/ml). Values are referred to the expression of _prf1_ in FB1 (wt) without pheromone. Each column represents the mean value of three independent biological replicates. Error bars represent the SD. The promoter from _tef1_ (encoding the translation elongation factor 1), which directs the expression of _prf1_ in the _prf1$^{con}$_ allele, is positively activated by TORC1 pathway, explaining the much higher expression of _prf1_ in the presence of Rheb-activating mutations.
(TIF)

**S17 Fig. Constitutive expression of _b_ genes alleviates the _rheb_-induced virulence defects.** (A) Strains derived from HA103 (expressing constitutively an active b heterodimer) and carrying the Rheb-activating alleles were spotted on minimal ammonium media charcoal plates and incubated for 24 h at 28˚C. The white fuzzy colonies reflect the formation of _b_-dependent

filaments. (B) Graph showing disease symptoms caused by infection with the indicated mutant strains. The symptoms were scored 14 days after infection. Three independent experiments were carried out, and the average values are expressed as a percentage of the total number of infected plants (n: 30 plants in each experiment).
(TIF)

**S18 Fig. Wide-field image sources from Fig 9B amplified images.** Bar: 15 μm.
(TIF)

**S19 Fig. Loss-of-function of Rsp5 and Art3 suppresses the inhibition of the transcriptional pheromone response resulted from the presence of Rheb-activating mutations.** qRT-PCR for the indicated strains (control, FB1 cells) incubated for 6 hours in minimal ammonium media in presence of synthetic a2 pheromone (2.5 μg/ml). Values are referred to the expression of each gene in control strain. Each column represents the mean value of three independent biological replicates. Error bars represent the SD.
(TIF)

**S20 Fig. Rsp5 mutants are avirulent.** (A) Graph showing disease symptoms caused by crosses of wild-type and mutant strains. The symptoms were scored 14 days after infection. Three independent experiments were carried out and the average values are expressed as percentage of the total number of infected plants (n: 30 plants in each experiment). (B) Representative leave 14 days after infection with the *rsp5Δ* strains.
(TIF)

**S21 Fig. Graphical abstract.**
(TIF)

**S1 Table. List of TOR components from *U. maydis* studied in this work.**
(DOCX)

**S2 Table. List of strains used in this study.**
(DOCX)

**S3 Table. List of qRT-PCR primers used in this study.**
(DOCX)

**S1 Methods. Sequence of DNA fragment for Gibson assembly to generate pCas9Rhb1KR (guide template in red).**
(DOCX)

**S2 Methods. Sequence of repair template carrying the rhebKR mutation (KR substitution in red, PAM mutation in blue, capital letters indicate coding sequence).**
(DOCX)

**S1 Source data. Source data.**
(XLSX)

## Acknowledgments

We thank Prof. J. Correa-Bordes (UNEX, Spain), Prof. A. di Pietro (UCO, Spain), and members of the Pérez-Martín lab for the discussion and critical reading of the manuscript. We also thank Prof. P. Vera (IBMCP, Valencia) and his laboratory for hosting us at the IBMCP.

## Author Contributions

**Conceptualization:** Antonio de la Torre, José Pérez-Martín.

**Formal analysis:** Antonio de la Torre.

**Funding acquisition:** José Pérez-Martín.

**Investigation:** Antonio de la Torre, José Pérez-Martín.

**Methodology:** Antonio de la Torre, José Pérez-Martín.

**Project administration:** José Pérez-Martín.

**Resources:** José Pérez-Martín.

**Supervision:** José Pérez-Martín.

**Writing – original draft:** José Pérez-Martín.

**Writing – review & editing:** Antonio de la Torre, José Pérez-Martín.

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
