## [Decision Letter · Decision Letter 0]

5 Jul 2022

Dear Dr Pérez-Martín,

Thank you very much for submitting your Research Article entitled 'The Rheb GTPase represses virulence by promoting  pheromone blindness via a TORC1-independent pathway in the phytopathogenic fungus Ustilago maydis' to PLOS Genetics.

The manuscript was fully evaluated at the editorial level and by independent peer reviewers. The reviewers appreciated the attention to an important topic, but raised some substantial concerns about the current manuscript. Based on the reviews, we will not be able to accept this version of the manuscript, but we would be willing to review a revised version. We cannot, of course, promise publication at that time.

If you decide to revise the manuscript for further consideration at PLOS Genetics, please aim to resubmit within the next 60 days, unless it will take extra time to address the concerns of the reviewers, in which case we would appreciate an expected resubmission date by email to plosgenetics@plos.org.

[LINK]

We are sorry that we cannot be more positive about your manuscript at this stage. Please do not hesitate to contact us if you have any concerns or questions.

Yours sincerely,

Eva H. Stukenbrock, PhD

Associate Editor

PLOS Genetics

Gregory P. Copenhaver

Editor-in-Chief

PLOS Genetics

Three reviewers provide below their detailed comments to the manuscript. These should all be considered in a revised and re-submitted version of the manuscript. This includes improved clarity in the presentation of the results. A summary figure with a schematic illustration of signaling branches would be helpful. Moreover, reviewer 1 points to sections in the results that clearly can be shortened while other details related to Rhb1KR mediated avirulence could be more extended.

The authors should also carefully take into account comments regarding missing controls. Although the reviewers acknowledge the use of different controls to support findings from the genetic analyses, some basic controls are missing in some of the experimental assays (IP, Western blot).

Several gene strains were generated, but the authors have not mentioned if the mutants were impaired in growth or in other ways showed a distinct phenotype. Reviewer 2 and 3 both highlight the necessity of providing information to mutant phenotypes as well as considering the possibility that deletion mutants could have accumulated suppressor mutations that complement gene loss.

Reviewer's Responses to Questions

**Comments to the Authors:**

Reviewer #1: In the manuscript by de la Torre and Perez-Martin, the authors investigate the relationship between TOR pathway activity and the virulence program in the corn smut fungus Ustilago maydis. The authors observed that Tor1 inhibition inhibits growth and components of the TORC1 but not TORC2 complexes are essential for growth in U. maydis. The further established Maf1 and Aga1 phosphorylation as potential read-out for TORC1 activity, respectively. To investigate the effects of increased TORC1 activity, the authors express a constitutive active alle of the Rheb GTPase Rhb1 or delete the Rhb1 GAP Tsc2. Rhb1KR expression results in aberrant localization and, most likely degradation of the nitrate transporter Nrt1, which is rescued upon rapamycin treatment, or deletion of the Rsp5 ubiquitin-ligase encoding gene (function downstream of Rhb1). Furthermore, in Rhb1KR expressing strains the virulence program is non-functional and filamentous growth during pheromone perception and formation of the infectious filament in haploid pathogenic background is blocked. Importantly, this phenotype appears to be independent of TORC1 activity as rapamycin treatment did not rescue the pheromone response. As a possible explanation the authors find that localization of the pheromone receptor Pra1 is disturbed upon Rhb1KR expression, which is rescued upon deletion of rsp5 or art3. In case of art3 deletion also rescues, in part the virulence defect of Rhb1KR expressing strains.

In conclusion, this manuscript provides first insight into TOR signaling in U. maydis, but focusses in its main part on the Tor-independent effect of Rhb1 overactivation on the virulence program. The underlying mechanism of Rhb1KR-dependent avirulence might relate to delocalized Pra1 but is not fully explored, understood or experimentally tested. This might also relate to various other reasons, as increased Rhb1 activity results in pleiotropic events.

Overall, the manuscript reports on several interesting observations, but needs to be overworked and restructured to improve coherence and focus. Several findings, appear to be unrelated to the main part of the paper. And based on the pleiotropic nature of many introduced mutations, conclusions are in some cases speculative.

Specific points:

1. To allow the reader appreciate the central and important findings of the manuscript some parts relating to Figures 1 + 2 + 4 should be more concise and the Rhb1KR mediated avirulence more expanded. Most of the observations in the first sections (page 6-10) conclude with knowledge that is expected and conserved and thus should be clearly shortened. (e.g. that TOR signaling is involved in sensing of nutritional cues..)

2. Please be more concise when introducing the relevant components of the TOR pathway in the individual sections and/or include, when necessary, in the introduction.

3. It would be important to find out if Pra1 delocalization is causal for the loss of virulence in Rhb1KR expressing strains. This hypothesis could be challenged by expression of prf1-con in the haploid pathogenic SG200 or HA103 background (together with Rhb1RK), as this de-necessitates the pheromone response pathway for virulence.

4. Please restructure and reword for clarity, grammar, conciseness and correct use of tenses. This is formally a minor point, but based on the numerous occasions these corrections are required to follow the logical flow of experiments.

5. To me, the title of the paper is somewhat misleading and not fully supported by the experiments. Please rephrase. Based on the data one can conclude that constitutive Rhb1 activity interferes with mating, the pheromone response and formation of infectious filaments. As SG200 is as well affected in virulence and filamentous growth, the Pra1 localization effect is unlikely to be the (only) cause for reduced virulence.

The conclusion that if filament formation is reduced in SG200, this shows that there is no defect in cell fusion in compatible matings is not justified (page 15). Rather it appears that both morphological transitions are not functioning.

Minor points:

1. Please clarify why the effect of Rhb1KR expression on Nrt1 functionality is relevant for the manuscript. In Fig 4E it is referred to RHB1KR-depndent degradation of Nrt1-GFP, although, overall abundance is clearly reduced. As this might as well result from altered translation, the degradation hypothesis is speculative and/or should be tested by further assays (e.g. cycloheximide treatment) or rephrased.

2. The authors should comment on why their protocols deviate from established U. maydis assays. For instance, precultures for virulence assays are normally done in YEPS medium, pheromone response is normally tested in CM and mating assay on PD charcoal plates. Here the authors use minimal media with ammonium as nitrogen source. It would be important to understand the underlying logic.

3. Formation of conjugation tubes and Fuz7DD filaments should be quantified.

4. Page 16: The main factor determining Prf1 activity is the differential phosphorylation by Kpp2 and PKA and not the mRNA level, which rather serves as an additional boost for signaling. Please clarify.

5. Page 17 (also on page 18): The suppression of Rhb1KR effects by Fuz7DD expression is not really convincing, as the morphological alterations upon Fuz7DD expression in WT and Rhb1KR cells are clearly different and do not resemble conjugation tubes. It is hard to see filamentous growth at all.

6. I was wondering if the authors could comment on the possibility whether Rhb1KR expression might interfere with cell cycle arrest, as this would explain the majority of the observed phenotypes (absence on conjugation tubes, infectious filaments, virulence).

7. In some instances rsp5 is misspelled rps5 in the manuscript.

8. In the discussion it is supposed that TORC1 activity is required to form infective filaments, although no data is presented in this respect. Rather the data indicates that high TORC1 activity might be detrimental for virulence.

Reviewer #2: Summary

In frame with previous discoveries, emphasizing the requirement of an obligate inhibition of Tor-signalling for efficient virulence in fungal pathogens, the work conducted by Antonio de la Torre and José Pérez-Martín aimed to further unravel Tor-signalling networks during virulence but also during vegetative growth in the phytopathogen, Ustilago maydis. Because very little is known about Tor-signalling in the vast majority of fungal pathogens, the authors had as prerequisite to assess the existence of TORC components and Tor-signalling, which they remarkably conducted. Besides putting impressive efforts that lead them i) to assess the essentiality of the Tor kinase, ii) to identify experimentally critical components of both TORC1 and TORC2 subunits as well as iii) TORC1 downstream effectors (Sch9 &Aga1 as direct effectors), they also remarkably identified Tsc2 and Rheb as TORC1 upstream regulators and used several genetic approaches, including deletant strains (for Tsc2), conditional alleles and hyperactive mutants, to assess their function as upstream regulators of TORC1. This contrasts with previous studies that mainly addressed Tor-regulation in pathogenic fungi through indirect approaches (i. e. modulating the fungal growth environment). As a major achievement, the authors have elegantly demonstrated that overactivation of Rheb1 hinders poor nitrogen source uptake through internalization and Rsp5-mediated degradation of the nitrate permease Nrt1 in a TORC1-dependent manner. With this in hand, Antonio de la Torre and José Pérez-Martín then focused on their initial problematic (impact of TORC1 overactivation in the pathogenic context) and showed that overactivity of Rheb1 abolishes the ability of U. maydis to form infective structures and thus their ability to infect plants. This was then shown to be due to the pheromone insensitivity, caused by misdistribution of the receptor Pra1, thereby hindering U. maydis to trigger the virulence program. Interestingly, the authors have shown that the inappropriate misdistribution of Pra1 in the virulence context also involves Rsp5 ubiquitin-ligase, as for the TORC1-dependent Ntr1 mis-signalisation previously demonstrated in this work. Strikingly, the Rheb1-mediated pheromone blindness during virulence was Tor-independent which was unexpected. It is noteworthy that the authors have managed to describe two Rheb1 and Rps5-dependent pathways both regulating external cues modulators of which only one is TORC1-dependent while the other is not.

Significance/originality

The proposed work will be of interest for a broad readership, including the fungi community but also people from the Tor-signalling field, notably because the present work uncovered Tsc-Rheb(-TORC1) dependent pathways that are not or poorly understood even in well-studied organisms. One out of the many strengths of the proposed work is the impressive effort the authors have put to evidence TOR-signalling in Ustilago maydis. It is highly appreciable that the authors dug into the TOR field in filamentous fungi, a field that is dramatically lacking scientific knowledge. It is easily expectable that lessons from this work are transposable to other fungi and will provide important perspectives for future research topics in fungal biology. I would like to point out the fact that the authors have unexpectedly but successfully deleted several genes that were expected to be essential (such as Rct, Sin1, Sch9 or Aga1). The significance of this research obviously also arises from the description of a critical regulation axis for virulence, and thereby evidences potential specific targets for the development of chemicals.

Quality of the manuscript

The manuscript is dense with information and data with very well thought out and carefully planned experiments and controls. The authors have carefully considered alternatives and addressed them with control experiments in nearly all cases. Their conclusions are clearly supported by the data presented. The manuscript is well-written, clear, easy to follow, and the conclusions are well justified. They provide a clear and complete introduction that will be appreciable for readers that are out of the higher fungi biology field. I also appreciated that all along the manuscript, the authors lead the readers by clearly establishing their aims, their issues, how they circumvent these issues and technical details about fungal biology or about the strategies they have employed. This is also appreciable in the figures within the main manuscript and in the Supplementary Material.

As a general observation, besides the below-mentioned remarks (mainly concerning blots), the figures are of high quality, designed in a sober way which greatly facilitates the reader to catch the author’s message. I also appreciated the way the authors have organized the figures between the main manuscript and the supplementary materials.

Remarks

• A main strategy used by the authors was to introduce an ectopic version of Rheb-K127R under nar1 promoter (overexpressed in the presence of nitrate (MMD) but not YPD). My understanding is that the endogenous Rheb might be inhibited on MMD media leading to TORC1 inhibition due to the poor nitrogen source. At the same time, the ectopic Rheb-K127R might be overexpressed leading to active TORC1 which is expected to induce Ntr1 degradation. Consequently, the cell should have defects in nitrate uptake which would lead to a negative feedback loop on the nar1 promoter. Nevertheless, the authors have shown that in spite of this potential negative feedback, the ectopic Rheb-K127R is overexpressed. This is surprising but can be explained due to the presence of alternative nitrate permeases.

My concern here is about the appropriateness of using a nitrate dependent promoter when aiming to unravel mechanistic insight involving nitrate uptake (i.e. via the Ntr1 permease). This point is according to me highly significant, and the authors should provide strong arguments defending why this strategy is still appropriate (or alternatively provide further data to circumvent this issue).

• The authors have conducted several immunoprecipitation (IP) experiments and evidenced changes in the phosphorylation status as readouts for TORC1-dependent and -independent pathways. They applied this approach in several cases by taking advantage from band shifts using SDS-PAGE conducted with phostag-gels. Although their results are convincing per se, I was somewhat surprised by the systematic lack of evident controls that must be done when conducting IP experiments. These include a blot showing the inputs that were submitted for IPs.

• Several blots are overexposed which hinders the evaluation of potential changes in protein amounts or potential double-bands (not applicable for blots evidencing on the same blots distinct bands on a same lane with drastic intensity variations such as free GFP accumulation versus full-length protein tagged with GFP, (for example Fig. 4C). Nevertheless, the authors could in this case cut the blot or display two expositions).

• Fig 4C: Loading control is present but no quantification. Because the loading is not homogenous, it is difficult to fully exploit this blot without any quantification. This is even more true because of that the loading control signal is somewhat saturated. Especially, I am pointing this out because the protein amount of Rhb1 wt + rps5 deletant strain on MMD (control lane 4) is clearly lower than the 3 other control conditions (when comparing Tubulin amounts). Thus, it is expectable that Nrt1-GFP level is comparable in the Rhb1 wt + rps5 deletant strain (if not higher) to the control strain on MMD. This would even further support the authors’ expectations. Quantifications would also facilitate the confrontation between protein (4C) and mRNA (4D) levels.

• The authors have successfully deleted several genes, including some of them that were expected to be essential. One might question whether these deletant strains have not accumulated suppressor mutations that complement the loss of the targeted gene? This point is not clearly discussed. How would the authors argue to this?

• In their work the authors attempted to demonstrate the essential character of several genes. To assess this, they took advantage of inducible/repressible promoters and convincingly reached their aims. However, the authors have recently successfully developed an Auxin-degron based approach in Ustilago maydis to create conditional null-mutants (through inducible targeted protein degradation; https://doi.org/10.1093/genetics/iyab152). I wondered why they didn’t take advantage from this system to assess the essentiality of the targeted genes.

• Fig1F: Despite the author attest that a similar amount of protein was loaded, the associated loading control is missing and there are evident differences in the Atg8-GFP signal. For example, in the second blot, lane 1 (MMD control) contains lower amount of Atg8-GFP than YPD torNAR. Is this due to differences in sample loading or to a physiological effect? This enables one to speculate that free GFP would also be present in the control strain on MMD if an equal amount was loaded. To avoid such issues, a loading control would be necessary. Also, the signals are saturated, and quantification is missing, even if I totally understand that the aim here is on the qualitative level rather than quantitative.

• Fig 8C: The blot is of low quality but sufficient to understand the authors’ message. For more clarity, I would suggest lowering “Pra1-GFP” mention in front of the expected band (67,63 kDa). Do the authors have an idea to what correspond the upper band at 250 kDa? Here I would also suggest to kindly remind the readers that defect in virulence due to variations in Pra1 levels was excluded as shown in figure 7B.

• The authors showed that while rapamycin has a mild effect on growth in U. maydis (compared to torin1; Supplementary Figure 3.) it was sufficient to restore all phenotypes observed under Tsc-Rheb-TORC1 overactivation (abolishing Aga1 phosphorylation (Figure 2B), ii) restoring the PM-localisation of Nrt1 in a rhb1-K127R (induced expression on MMD; Fig 4A) as well as iii) restoring its protein level (Figure 4C)). The authors concluded that these pathways are TORC1 dependent. Because of the slight inhibitory effect of Rapamycin on growth, could the authors envisage to delete the proline isomerase FKBP12 homolog in U. maydis? In this context, Rapamycin treatment might not induce the above-mentioned events which would further demonstrate their TORC1-dependencies.

• The authors reported Aga1 as being the closest homolog of YPKs (from S. cerevisiae) and Gad8 (from S. pombe). In contrast to YPK1/2 homologs, which are direct TORC2 targets, S. cerevisiae also encodes an additional YPK homolog (YPK3) which is a substrate of TORC1. Of note, once activated by TORC1, YPK3 phosphorylates Rps6. In their work, the authors showed that Aga1 deletion had only a mild impact on growth, then they showed it to be regulated by TORC1 and concluded it to be a substrate of TORC1. However, they didn’t discuss, nor have they tested, the possibility of Aga1 being a functional homolog of YPK3 rather than of YPK1/2. This arises the possibility that the TORC2 substrate (i. e. the functional homolog of YPK1/2) in U. maydis was not identified.

• I would also suggest changing the term S6K by AGC kinases to designate YPK and Sch9 homologs. In facts, S6K kinases are a sub-group of AGC kinases (doi:10.1038/nrm2822 for AGC classification) to which YPK do not belong (they belong to AGC/Akt kinases). Whether Aga1 and other kinases mentioned in the manuscript indeed belong to the Akt or to the S6k is not clear.

• The authors have obtained viable deletant strains of genes expected to encode specific components of TORC2. In spite those genes were not essential in U. maydis, I wondered whether the authors tested the impact of those deletions (i. e. rct and sin1 deletant) for virulence (for example, impact on the development of infective hyphae)? Conversely have they tested the impact of torin1 (inhibiting TORC1 & TORC2 in other systems) treatment during virulence?

• Page 16: There is a little error of inattention while choosing the terms: The authors reported “Elegant previous work had described that Prf1 is submitted to posttranscriptional control by protein phosphorylation… [65, 66]”. The herein cited references have indeed evidenced an posttranscriptional regulation of Prf1 but also a posttranslational regulation. I think that the authors meant “posttranslational control” instead of posttranscriptional since they refer to Prf1 phosphorylation.

• Primers table used in the work is missing and should be included

• Suggestion: If I might, although very clearly presented, readers might appreciate a schematic recapitulating both signalling branches (TSC-Rheb TORC1-dependent axis, leading to Rps5 mediated degradation of Ntr1, and TSC-Rheb TORC1-independent overactivation hindering virulence via pheromone receptor Pra1 mis-localisation. In this schematic, TSC-Rheb and rps5 could share the characteristic of being involved in TORC1 dep and indep paths). This schematic could even include the TORC1/(2) upstream and downstream components evidenced here.

Reviewer #3: The authors (de la Torre and Pérez-Martin) present an investigation of the role of TORC1 in the virulence of Ustilago maydis using a variety of genetic, pharmacological and biochemical approaches. They made the interesting discovery that a TORC1-independent pathway influences the virulence of the fungus by interfering with the response to pheromone. The main approach was to use a gain-of-function version of the Rheb GTPase, an upstream regulator of TOR, and to assess the roles of TOR-related functions, degradative functions for permeases and receptors, and components of the pheromone signaling pathway. The study is impressive because of the methodical and detailed evaluation of functions related to TORC1 activity, the pursuit of relevant side avenues (e.g., nitrate permease degradation), the use of regulated promoters to bypass essentiality, and the linkage with virulence via infection assays in maize. Overall, the data are clearly presented and interpreted. I have a few minor suggestions for improvements and clarification.

1.There are a number of situations where incorrect words are used. For example, the following sentence appears in the Abstract: “Strikingly, this negative control does not imply the TOR pathway.” The word imply is not used correctly. Other example words are: concourse, acuation, appropriated, subdued, coherent, seek, etc. I recommend that the authors proof read the entire manuscript to check work usage.

2.Page 7. Lst8 is briefly mentioned as a candidate gene to be deleted but the authors don’t follow up with information about whether the gene was successfully deleted and, if so, whether there was a relevant phenotype.

3. Fig. 3F and page 11. More explanation is needed for the statement: “This effect was more apparent in the rhb1K127R allele expression than upon tsc2 removal.”

4. It wasn’t clear why YPD was used as the restrictive condition for the nar1 promoter – the comparison with MMD as the permissive condition could potentially be confounded by other differences in media composition.

5. Figure 6B,C. Were different concentrations of rapamycin tested to ensure that an appropriate level was employed?

6. Figure 8C is quite messy and the GFP bands are highly overexposed. Perhaps the authors can comment on the reliability of their interpretations given the presentation.

7. Given the complexity of the study in terms of the number of genes/mutants/regulated strains employed, the authors might consider adding a summary diagram.

**Have all data underlying the figures and results presented in the manuscript been provided?**

Reviewer #1: Yes

Reviewer #2: Yes

Reviewer #3: Yes

PLOS authors have the option to publish the peer review history of their article (what does this mean?). If published, this will include your full peer review and any attached files.

Reviewer #1: No

Reviewer #2: No

Reviewer #3: No

---

## [Decision Letter · Decision Letter 1]

17 Oct 2022

Dear Dr Pérez-Martín,

We are pleased to inform you that your manuscript entitled "The Rheb GTPase promotes pheromone blindness via a TORC1-independent pathway in the phytopathogenic fungus Ustilago maydis" has been editorially accepted for publication in PLOS Genetics. Congratulations!

Yours sincerely,

Eva H. Stukenbrock, PhD

Section Editor

PLOS Genetics

Gregory Copenhaver

Editor-in-Chief

PLOS Genetics

Comments from the reviewers (if applicable):

Reviewer's Responses to Questions

**Comments to the Authors:**

Reviewer #1: The authors have done a great job and addressed all major issues raised regarding the previous submission. The manuscript is well suited for PLoS Genetics and I expect it will engage readers from various fields of molecular biology.

Reviewer #2: The authors have satisfyingly addressed all my remarks. The implemented material, whether it is about additional experiments or modifications of the manuscript, is of high quality and accurately answers all issues addressed to the authors. The initial version of the manuscript submitted by the authors was significantly improved and I have no other issues to address.

Reviewer #3: The authors adequately addressed my comments.

**Have all data underlying the figures and results presented in the manuscript been provided?**

Reviewer #1: Yes

Reviewer #2: Yes

Reviewer #3: Yes

PLOS authors have the option to publish the peer review history of their article (what does this mean?). If published, this will include your full peer review and any attached files.

Reviewer #1: No

Reviewer #2: No

Reviewer #3: No

**Data Deposition**

http://datadryad.org/submit?journalID=pgenetics&manu=PGENETICS-D-22-00617R1

**Press Queries**

---

## [Editor Report · Acceptance letter]

7 Nov 2022

PGENETICS-D-22-00617R1 

The Rheb GTPase promotes pheromone blindness via a TORC1-independent pathway in the phytopathogenic fungus Ustilago maydis 

Dear Dr Pérez-Martín, 

We are pleased to inform you that your manuscript entitled "The Rheb GTPase promotes pheromone blindness via a TORC1-independent pathway in the phytopathogenic fungus Ustilago maydis" has been formally accepted for publication in PLOS Genetics! Your manuscript is now with our production department and you will be notified of the publication date in due course.

With kind regards,

Zsofi Zombor

PLOS Genetics

On behalf of:
